# Neprilysins regulate muscle contraction and heart function via cleavage of SERCA-inhibitory micropeptides

Ronja Schiemann [1,6], Annika Buhr [1,6], Eva Cordes[1], Stefan Walter[2], Jürgen J. Heinisch [2,3], Paola Ferrero [4], Hendrik Milting[5], Achim Paululat [1,2] & Heiko Meyer [1,2] ✉

Muscle contraction depends on strictly controlled $Ca^{2+}$ transients within myocytes. A major player maintaining these transients is the sarcoplasmic/endoplasmic reticulum $Ca^{2+}$ ATPase, SERCA. Activity of SERCA is regulated by binding of micropeptides and impaired expression or function of these peptides results in cardiomyopathy. To date, it is not known how homeostasis or turnover of the micropeptides is regulated. Herein, we find that the *Drosophila* endopeptidase Neprilysin 4 hydrolyzes SERCA-inhibitory Sarcolamban peptides in membranes of the sarcoplasmic reticulum, thereby ensuring proper regulation of SERCA. Cleavage is necessary and sufficient to maintain homeostasis and function of the micropeptides. Analyses on human Neprilysin, sarcolipin, and ventricular cardiomyocytes indicates that the regulatory mechanism is evolutionarily conserved. By identifying a neprilysin as essential regulator of SERCA activity and $Ca^{2+}$ homeostasis in cardiomyocytes, these data contribute to a more comprehensive understanding of the complex mechanisms that control muscle contraction and heart function in health and disease.

The contraction of muscle fibers is governed by well-characterized molecular processes, with the concentration of free cytosolic calcium ions being a crucial parameter. This concentration is largely maintained by the activity of the sarcoplasmic and endoplasmic reticulum $Ca^{2+}$ ATPase (SERCA), an enzyme that transports $Ca^{2+}$ from the cytosol into the sarcoplasmic reticulum (SR). The resulting reduction in cytosolic $Ca^{2+}$ levels initiates the muscle relaxation phase. Accordingly, precise regulation of SERCA activity is essential for the proper function of muscle tissue, and the physiological relevance of SERCA in heart and muscle disease has been extensively studied (see[1,2] and references therein). One central result of these studies is that the amount and activity of SERCA are significantly reduced upon aging as well as under

pathophysiological conditions, such as congestive heart failure or progressive muscular dystrophy. In addition, an increase in SERCA activity ameliorates corresponding etiopathologies, which provides further evidence for the critical relevance of SERCA to muscle physiology. Moreover, these data emphasize the therapeutic potential of modulating SERCA activity in a directed manner[1,3–6].

A prerequisite for the development and implementation of appropriate therapies is knowledge of all relevant factors affecting SERCA activity. In this context, the high degree of structural and functional conservation between human and *Drosophila* SERCA renders the fly an ideal model system to identify relevant factors[7], especially since also in *Drosophila*, SERCA activity is essential to

[1]Department of Zoology & Developmental Biology, Osnabrück University, 49076 Osnabrück, Germany. [2]Center of Cellular Nanoanalytics Osnabrück - CellNanOs, 49076 Osnabrück, Germany. [3]Department of Genetics, Osnabrück University, 49076 Osnabrück, Germany. [4]Center for Cardiovascular Research - CONICET/National University of La Plata, 1900 La Plata, Argentina. [5]Heart & Diabetes Center NRW, University of Bochum, Erich & Hanna Klessmann-Institute for Cardiovascular Research and Development, 32545 Bad Oeynhausen, Germany. [6]These authors contributed equally: Ronja Schiemann, Annika Buhr. ✉e-mail: Meyer@biologie.uni-osnabrueck.de

proper cardiac function[8,9]. Both in vertebrates[10] and in *Drosophila melanogaster*[11], the activity of SERCA is controlled by certain SR membrane integral peptides that bind to the enzyme and inhibit its activity. In vertebrate hearts, Phospholamban (PLN) and Sarcolipin (SLN) have been identified as such regulatory peptides[12,13], with the affinity of PLN for SERCA depending on PLN phosphorylation and oligomerization states[10]. For both micropeptides, alterations in expression or function are associated with severe cardiomyopathy[14–19]. Recently, two additional micropeptides that bind and regulate SERCA in muscle tissue have been identified: myoregulin (MLN) and dwarf open reading frame (DWORF). MLN shares sequence and structural similarity with PLN and SLN and appears to be a major peptide inhibitor of SERCA activity in fast-type skeletal muscle[20]. By contrast, DWORF represents the only SERCA-activating micropeptide identified up to now[21,22]. Among these peptide regulators of SERCA activity, MLN, PLN, and SLN share several amino acids within their transmembrane helices that are critical to the interaction with SERCA[20]. These residues are also conserved in Sarcolamban A (SCLA) and Sarcolamban B (SCLB), invertebrate SERCA-inhibitory micropeptides present in cardiac and somatic muscles of *Drosophila melanogaster*[11,20]. Analogous to human PLN, SLN, and MLN, the binding of SCL to *Drosophila* SERCA reduces the activity of the enzyme. Accordingly, SCL loss-of-function mutants exhibit impaired calcium transients in heart cells, concomitant with heart arrhythmia[11]. To date, mechanisms that regulate the amount or the turnover of any of the SERCA-regulatory micropeptides are largely unknown.

In this study, we present evidence that the endopeptidase Neprilysin 4 (Nep4) acts as an essential regulator of SCL homeostasis and SERCA activity in *Drosophila*. Increased *nep4* expression in heart cells phenocopied characteristic effects of *scl* knockout, including impaired Ca²⁺ transients, aberrant SERCA activity, and heart arrhythmia. Respective phenotypes strictly depended on the catalytic activity of the enzyme, which identified abnormal peptide cleavage as a critical factor. In a combined in vitro and in vivo approach, we identified Sarcolamban peptides as the phenotype-relevant Nep4 substrate and could show that hydrolysis of the peptides reduces their membrane affinity, thereby releasing them from the SR membrane and precluding further SERCA interaction. In addition, the Nep4-hydrolyzed peptides exhibited a significantly impaired ability to oligomerize. Finally, Nep4, SERCA, and SCL colocalized in SR membranes of heart and body wall muscles, suggesting that spatial proximity between the three factors is of high mechanistic relevance. Initial analyses on corresponding human factors indicated that the regulatory mechanisms identified in *Drosophila* are relevant also in humans.

## Results

### Modulating *nep4* expression affects cardiac function, Ca²⁺ homeostasis, and SERCA activity

In a screen for candidate proteins regulating heart function in adult *Drosophila melanogaster*, we identified the metallopeptidase Nep4 as an essential factor. Using semi-automated optical heartbeat analysis (SOHA[23]), physiological heart parameters were analyzed in animals exhibiting either increased or reduced expression of the peptidase in heart cells. In addition to the active enzyme, the effects of overexpressing a catalytically inactive Nep4 variant, containing glutamine instead of the essential glutamate within the zinc-binding motif (E873Q[24,25]), were assessed (Fig. 1). Animals overexpressing active Nep4 displayed both arrhythmia and prolonged periods of diastolic heart arrest, while increased production of the catalytically inactive variant had no such effects. Cardiomyocyte-specific *nep4* knockdown resulted in a tendency toward arrhythmia, but the effect was not statistically significant (Fig. 1A, D and Supplementary Movies 1–4). However, compared to controls, abnormally long heart periods (HP) were recurrently observed in knockdown animals, with individual periods

lasting up to 4.500 ms (Fig. 1C and Supplementary Movie 4). Prolonged HPs were also characteristic for increased *nep4* expression, yet to an even stronger extent (individual HP > 20.000 ms, Fig. 1C and Supplementary Movie 2). Unlike rhythmicity, heart rate was not affected by increased or reduced *nep4* expression. Corresponding animals exhibited a median heart rate of about 90–100 beats per minute, which was not significantly different from control hearts (Fig. 1B) and within the range of published data[26,27]. Similar inter-individual Nep4 overexpression levels were confirmed by immunostainings (Supplementary Fig. 9A, B). To exclude nonspecific effects caused by the upstream activation sequence (UAS) constructs employed or owed to the chromosomal background, all analyzed lines (UAS-Nep4, UAS-Nep4^E873Q, and UAS-*nep4* RNAi) were crossed to *w*¹¹¹⁸ flies. None of the control offspring exhibited aberrant heart parameters (Fig. 1A, B).

To evaluate whether the functional impairments were specific to cardiomyocytes or affected muscle cells in general, we analyzed third instar larval body wall muscles for corresponding deficiencies. Endogenous *nep4* expression in cardiac and somatic muscle tissue of all *Drosophila* developmental stages has been confirmed previously[24,25,28,29]. Consistent with the effects in heart cells, body wall muscle function was significantly impaired by increased Nep4 levels. Relative to controls, corresponding animals exhibited a reduced crawling speed (Fig. 1E) that was based on a decreased muscle contraction frequency (Fig. 1F), while the magnitude of the individual contractions was not affected (Supplementary Fig. 9C). Again, the phenotype depended on the enzymatic activity of Nep4; increased expression of the catalytically inactive variant did not affect any of the measured parameters. RNAi-mediated knockdown of the peptidase was also without significant effects (Fig. 1E, F). Equivalent expression levels and wildtype-like subcellular localizations of all Nep4 overexpression constructs, as well as a high *nep4*-specific knockdown capacity of the analyzed RNAi line, were confirmed previously[24,25].

To assess the physiological basis of the observed phenotypes, we employed a genetically encoded Ca²⁺ indicator (GCaMP3) to analyze Ca²⁺ transients in cardiomyocytes exhibiting altered *nep4* expression. We found that both increasing and reducing *nep4* levels affected cardiac Ca²⁺ handling, namely SR Ca²⁺ load, SERCA activity, and cardiac relaxation constants (Fig. 2). To estimate the SR Ca²⁺ load, we analyzed caffeine-induced Ca²⁺ transients in the corresponding transgenic animals. Caffeine activates ryanodine receptors; thus, measuring caffeine-induced transients allows calculation of the Ca²⁺ content within the SR[30,31]. Interestingly, *nep4* overexpression animals exhibited a significant increase in SR Ca²⁺ load, while animals overproducing the catalytically inactive variant did not (Fig. 2A). Since a major factor determining the degree of the SR Ca²⁺ load is SERCA[2], we subsequently calculated the activity of this enzyme. In accordance with the increased Ca²⁺ load, overexpression of active Nep4 increased SERCA activity by about twofold. Vice versa, cardiac-specific *nep4* knockdown significantly reduced SERCA activity. Again, only the enzymatically active Nep4 affected SERCA activity, while catalytically inactive Nep4 had no such effect (Fig. 2B). Finally, we determined the constant of relaxation (Tau) in the respective transgenic animals (Fig. 2C). In agreement with reduced SERCA activity, *nep4* knockdown caused a significant increase in Tau, as reflected by prolonged relaxation in corresponding Ca²⁺ traces (Fig. 2D). Conversely, overexpression of catalytically active Nep4 resulted in a tendency toward a reduced Tau. However, the effect was not statistically significant (Fig. 2C). To exclude any influence of variable beating frequencies on the measured Ca²⁺ flux parameters, we analyzed a subgroup of *nep4* knockdown flies selected specifically for slower heart rates. For this group, all effects on SR Ca²⁺ load, SERCA activity, and Tau were consistent with the population data of that genotype (Supplementary Table 2 and Fig. 2A–C), indicating variations in heart rate did not significantly affect the measured parameters. Equivalent indicator concentrations in cardiomyocytes of all

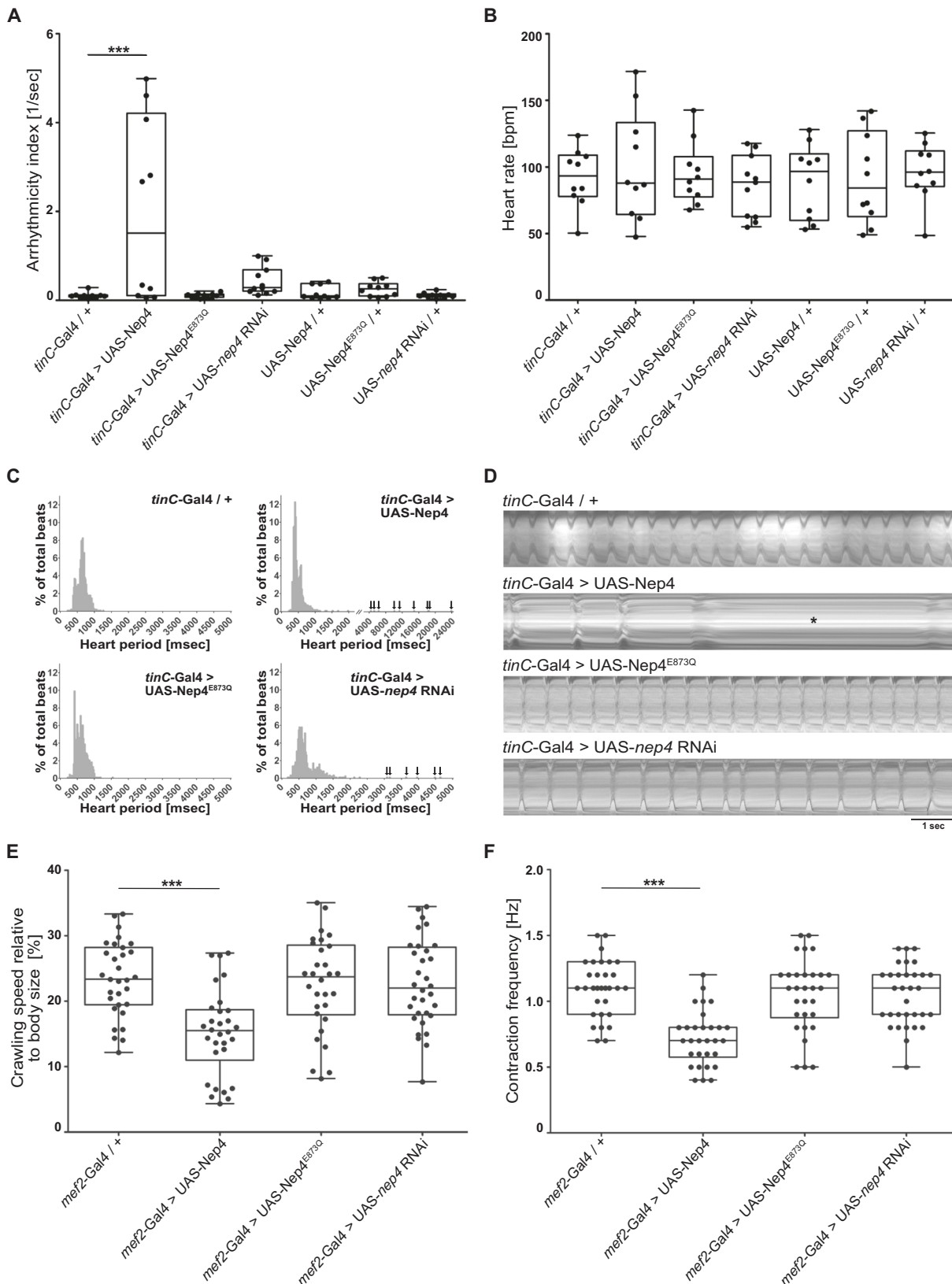

transgenic lines tested were confirmed by Western blot-based single-animal specific quantifications of GCaMP3 protein amounts (Supplementary Fig. 9E). The possibility that the observed changes in SERCA activity were based on altered expression of this enzyme was assessed by Western blots using protein extracts isolated from animals of all genotypes tested. SERCA expression levels and protein stability were not affected by altered *nep4* expression (Fig. 2E), indicating a direct regulatory effect of Nep4 on the activity of SERCA.

## Nep4 colocalizes and coprecipitates with SERCA

To understand the Nep4-dependent effects on myocytic Ca²⁺ homeostasis in more detail, the subcellular localization of Nep4 in adult

**Fig. 1 | Nep4 activity is critical to heart and body wall muscle function.**
**A** Relative to control animals (*tinC*-Gal4/+, $n = 10$), cardiomyocyte-specific over-expression of catalytically active Nep4 (*tinC*-Gal4>UAS-Nep4, $n = 10$) causes arrhythmia. Overexpression of inactive Nep4 (*tinC*-Gal4>UAS-Nep4[E873Q], $n = 10$) or reduced expression of the peptidase (*tinC*-Gal4>UAS-*nep4* RNAi, $n = 11$) do not affect rhythmicity. UAS controls (UAS-Nep4, $n = 10$; UAS-Nep4[E873Q], $n = 10$; UAS-*nep4* RNAi, $n = 10$) are also without any effect. **B** Relative to control animals (*tinC*-Gal4/+, $n = 10$), neither knockdown of *nep4* (*tinC*-Gal4>UAS-*nep4* RNAi, $n = 11$) nor increased expression of active (*tinC*-Gal4>UAS-Nep4, $n = 10$) or inactive Nep4 (*tinC*-Gal4>UAS-Nep4[E873Q], $n = 10$) affect heart rate. **C** Combined histograms showing the distribution of heart periods (HP) from flies of the indicated genotypes. Over-expression of catalytically active Nep4 (*tinC*-Gal4>UAS-Nep4, $n = 10$, 964 recorded beats) as well as reduced expression of the peptidase (*tinC*-Gal4>UAS-*nep4* RNAi, $n = 11$, 881 recorded beats) result in the occurrence of abnormally long HPs (arrows). Control hearts (*tinC*-Gal4/+, $n = 10$, 897 recorded beats) or hearts over-expressing inactive Nep4 (*tinC*-Gal4>UAS-Nep4[E873Q], $n = 10$, 929 recorded beats) do not exhibit such impairments. **D** Representative 10 s M-mode traces depict heart contractions from flies of the indicated genotypes. Increased expression of catalytically active Nep4 (*tinC*-Gal4>UAS-Nep4) causes arrhythmia with prolonged periods of diastolic heart arrest (asterisk). **E** Relative to control animals (*mef2*-Gal4/+, $n = 31$), muscle-specific Nep4 overexpression (*mef2*-Gal4>UAS-Nep4, $n = 30$) decreases crawling speed by 34%. Increased expression of catalytically inactive Nep4 (*mef2*-Gal4>UAS-Nep4[E873Q], $n = 30$) or knockdown of the peptidase (*mef2*-Gal4>UAS-*nep4* RNAi, $n = 32$) has no effect. **F** Relative to control animals (*mef2*-Gal4/+, median: 1.1 Hz, $n = 31$), muscle-specific Nep4 overexpression (*mef2*-Gal4>UAS-Nep4, $n = 30$) decreases muscle contraction frequency (median: 0.7 Hz). Increased expression of catalytically inactive Nep4 (*mef2*-Gal4>UAS-Nep4[E873Q], $n = 30$) or knockdown of the peptidase (*mef2*-Gal4>UAS-*nep4* RNAi, $n = 32$) has no significant effect (median: 1.1 and 1.1 Hz, respectively). Asterisks indicate statistically significant deviations from respective controls ($p < 0.001$, one-way ANOVA followed by Dunnett's Multiple Comparison Test). For all experiments, at least 10 individual animals were analyzed per genotype. Source data are provided as a Source data file.

cardiomyocytes and heart-associated ventral longitudinal muscles[32] was analyzed. We employed the previously established endogenous *nep4* enhancer[28] to drive the expression of HA-tagged *nep4A* gene product (Nep4A::HA) at near-physiological levels. Identical subcellular localizations of endogenous and tagged versions of the protein were confirmed previously[24]. As depicted in Fig. 3, within the heart and ventral longitudinal muscles, Nep4 localized to endomembranes, including the nuclear membrane (Fig. 3A´, C´, D´, F´, arrowheads). In addition, a punctate repetitive pattern along the muscle fibers was apparent (Fig. 3A´, C´, D´, F´, arrows). Identically-treated control crosses lacking the UAS construct (*nep4 > w[1118]*) did not exhibit any signal above background (Fig. 3G, G´). Co-labeling using anti-SERCA antibodies resulted in considerable signal overlap, indicating colocalization of the two enzymes within membranes of the SR compartment (Fig. 3B, C´, E–F´). A similar pattern was evident in larval heart and body wall muscles (Supplementary Fig. 1). This observation, together with the consistent responses to modified *nep4* expression in cardiomyocytes and body wall muscles (Fig. 1), suggests analogous functions of the peptidase in the two contractile tissues and across developmental stages.

To assess whether colocalization of Nep4 and SERCA reflected an interaction at the protein level, we performed pull-down assays using a green fluorescent protein (GFP) tagged Nep4 fusion protein (Nep4::GFP) as bait (Fig. 3I). The construct was expressed in muscles (*mef2*-Gal4) to probe interaction in tissue with endogenous relevance of SERCA and Nep4. SERCA efficiently coprecipitated with Nep4::GFP, indicating physical interaction between the two enzymes. Free cytoplasmic GFP (*mef2*-Gal4>GFP, Fig. 3I), as well as SR-luminal GFP (*mef2*-Gal4 > GFP.ER, Supplementary Fig. 9D) were used as individual controls. Coprecipitation of SERCA was absent in either control.

## Nep4 colocalizes with SERCA-inhibitory Sarcolamban peptides
As depicted in Fig. 1, all Nep4-mediated effects on cardiac Ca[2+] homeostasis and SERCA activity depended on the catalytic activity of the peptidase. This result implies that aberrant hydrolysis of certain peptides represents the mechanistic basis of the observed phenotypes. To date, only two SCL peptides have been identified as peptidergic regulators of SERCA activity in *Drosophila*[11]. We found that Nep4::GFP and FH-tagged SCLA (FH::SCLA) partially colocalized with SERCA in the body wall (Supplementary Fig. 3) as well as in cardiac muscle cells (Fig. 4A–A´´´, B). In cardiomyocytes, substantial colocalization of all three factors was observed around the nucleus (Fig. 4B, solid arrow). More distant from the nucleus, mainly SERCA and SCLA signals were visible (Fig. 4B, open arrows). A similar localization pattern was observed for SCLB, Nep4, and SERCA (Supplementary Fig. 2). Correct functionality and localization of the FH-tagged constructs had been confirmed previously[11]. To analyze the subcellular localization in more detail, we performed stimulated

emission depletion (STED)-based super-resolution imaging of corresponding heart preparations (Fig. 4C–C´´´, D, D´). The results indicated a rather dynamic local distribution of the three factors, with considerable overlap of Nep4, SERCA, and SCLA signals being evident in membranes contiguous with the nuclear membrane (Fig. 4D, solid arrow). In addition, individual colocalization between SCLA and SERCA (Fig. 4D, open arrow), Nep4 and SERCA (Fig. 4D, feathered arrow), or Nep4 and SCLA (Fig. 4D, solid winged arrowhead) was occasionally observed. Finally, we also detected free SERCA (Fig. 4D, open arrowhead), Nep4 (Fig. 4D, open winged arrowhead), and SCLA (Fig. 4D, solid arrowhead). More distant from the nucleus, mainly SERCA and SCLA were detected (Fig. 4D´). Again, a colocalizing portion (Fig. 4D´, open arrow), as well as individual SERCA (Fig. 4D´, open arrowhead) and SCLA signals (Fig. 4D´, solid arrowhead), were present.

## Neprilysins hydrolyze the luminal domain of SERCA-inhibitory peptides from flies and humans
Based on their spatial proximity within the SR membrane, we analyzed whether SCL represents a substrate of Nep4. To narrow down possible cleavage sites, we employed a Nep4::roGFP fusion protein to determine whether the catalytic center of Nep4 and thus its hydrolytic activity, is oriented towards the SR lumen or the cytosol of muscle cells. As shown in a previous study, the excitation spectrum of roGFP is highly redox sensitive[33,34]. Accordingly, the steady-state fluorescence intensity ratio (405/488 nm excitation) of the free roGFP control reflected the reducing conditions present in the cytoplasm (Supplementary Fig. 4A). By contrast, roGFP fused to the C-terminus of Nep4 exhibited a steady-state fluorescence intensity ratio characteristic of oxidizing conditions (Supplementary Fig. 4B). This indicates that the C-terminus of Nep4, and thus its catalytic activity, resides in the SR lumen, suggesting that Nep4-mediated hydrolysis of target peptides occurs within this compartment. Of note, most of the SERCA-regulatory peptides identified to date, including the *Drosophila* SCL, harbor a luminal domain at their C-terminus[20]. For human PLN, a single point mutation within this domain (V49A) completely disrupts the inhibitory effects of PLN on SERCA2a[12], which demonstrates the physiological relevance of the luminal domain.

To analyze whether the Nep4-mediated regulation of SERCA activity involves cleavage of the corresponding SCL peptide domains, we assessed the susceptibility of the luminal termini of SCLA (YLIYAVL, SCLA[lum]) and SCLB (YAFYEAAF, SCLB[lum]) to Nep4-mediated hydrolysis. To this end, recombinant Nep4 was purified from transfected *Sf*21 insect cells and incubated with the respective peptides. As depicted in Fig. 5, Nep4 hydrolyzed both SCLA[lum] and SCLB[lum] at distinct positions. While SCLA[lum] was mainly cleaved between Ala-5 and Val-6 (Fig. 5A, red chromatogram), SCLB[lum] was hydrolyzed predominantly between Ala-2 and Phe-3 as well as

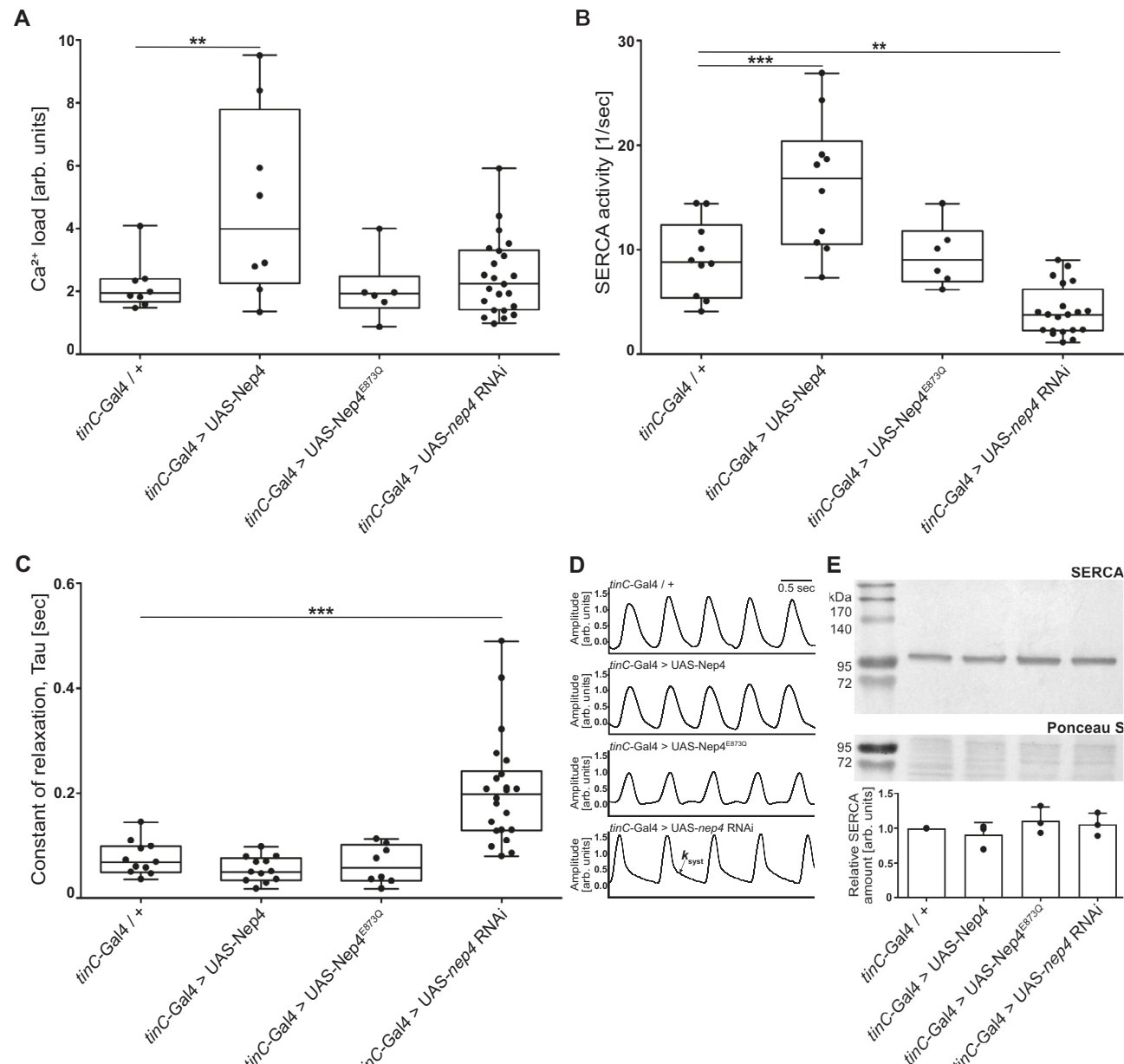

**Fig. 2 | Cardiac Ca²⁺ homeostasis is affected by altered nep4 expression levels.**
**A** Relative to control animals (*tinC*-Gal4/+, $n = 8$), sarcoplasmic reticulum Ca²⁺ load is increased by 105% in cardiomyocytes overexpressing active Nep4 (*tinC*-Gal4 > UAS-Nep4, $n = 8$). Overexpression of catalytically inactive Nep4 (*tinC*-Gal4 > UAS-Nep4^E873Q, $n = 6$) or knockdown of the peptidase (*tinC*-Gal4 > UAS-*nep4* RNAi, $n = 23$) has no significant effect. **B** Relative to control animals (*tinC*-Gal4/+ $n = 10$), over-expression of active Nep4 (*tinC*-Gal4 > UAS-Nep4, $n = 10$) results in an increase in SERCA activity by 91%, while knockdown of the peptidase (*tinC*-Gal4 > UAS-*nep4* RNAi, $n = 20$) reduces SERCA activity by 57%. Overexpression of catalytically inac-tive Nep4 (*tinC*-Gal4 > UAS-Nep4^E873Q, $n = 6$) has no significant effect. **C** The constant of relaxation (Tau) is only affected by the knockdown of *nep4*. Relative to controls (*tinC*-Gal4/+, $n = 11$), corresponding animals (*tinC*-Gal4 > UAS-*nep4* RNAi, $n = 22$) exhibit a 2.9-fold increase in Tau. Overexpression of active (*tinC*-Gal4 > UAS-Nep4,

$n = 12$) or inactive Nep4 (*tinC*-Gal4 > UAS-Nep4^E873Q, $n = 8$) has no significant effects. Asterisks indicate statistically significant deviations from respective controls (**$p < 0.01$, ***$p < 0.001$, one-way ANOVA followed by Dunnett's Multiple Compar-ison Test). Each dot represents one analyzed animal. **D** Representative Ca²⁺ traces indicating the decay rate constant of the systolic Ca²⁺ transients ($k_{syst}$). Traces from 1-week-old adult *Drosophila* hearts of the depicted genotypes are shown.
**E** Representative Western blot of total protein extracts isolated from adult flies of the indicated genotypes. For quantification, pixel intensity measurements were normalized to corresponding loading controls (Ponceau S). The resulting values are shown relative to the control (*tinC*-Gal4/+). The lower panel depicts the mean values (+SD) of three individual biological replicates. SERCA protein levels are not affected by altered *nep4* expression (paired *t* test, two-tailed). Source data are provided as a Source data file.

---

between Ala-7 and Phe-8 (Fig. 5B, red chromatogram). Identically-treated control preparations lacking Nep4 did not exhibit any hydrolytic activity (Fig. 5A, B, green chromatograms). The observed cleavage specificities agree with the reported preference of Nep4 to cleave next to hydrophobic residues, particularly with Phe or Leu at P1′[24]. To confirm the mechanistic relevance of the amino acids at P1′, and thus the specificity of the Nep4-mediated SCL cleavage, we also

analyzed mutated forms of the respective peptides (SCLA_lum(V6A); SCLB_lum(F8A), SCLB_lum(F3A), and SCLB_lum(F3A/F8A), Supplementary Fig. 5). In all cases, the substitution of the P1′ residue by Ala resulted in altered cleavage characteristics, ranging from the generation of novel hydrolysis products (Supplementary Fig. 5A), up to a complete resistance of the corresponding derivative to Nep4 mediated hydrolysis (Supplementary Fig. 5D).

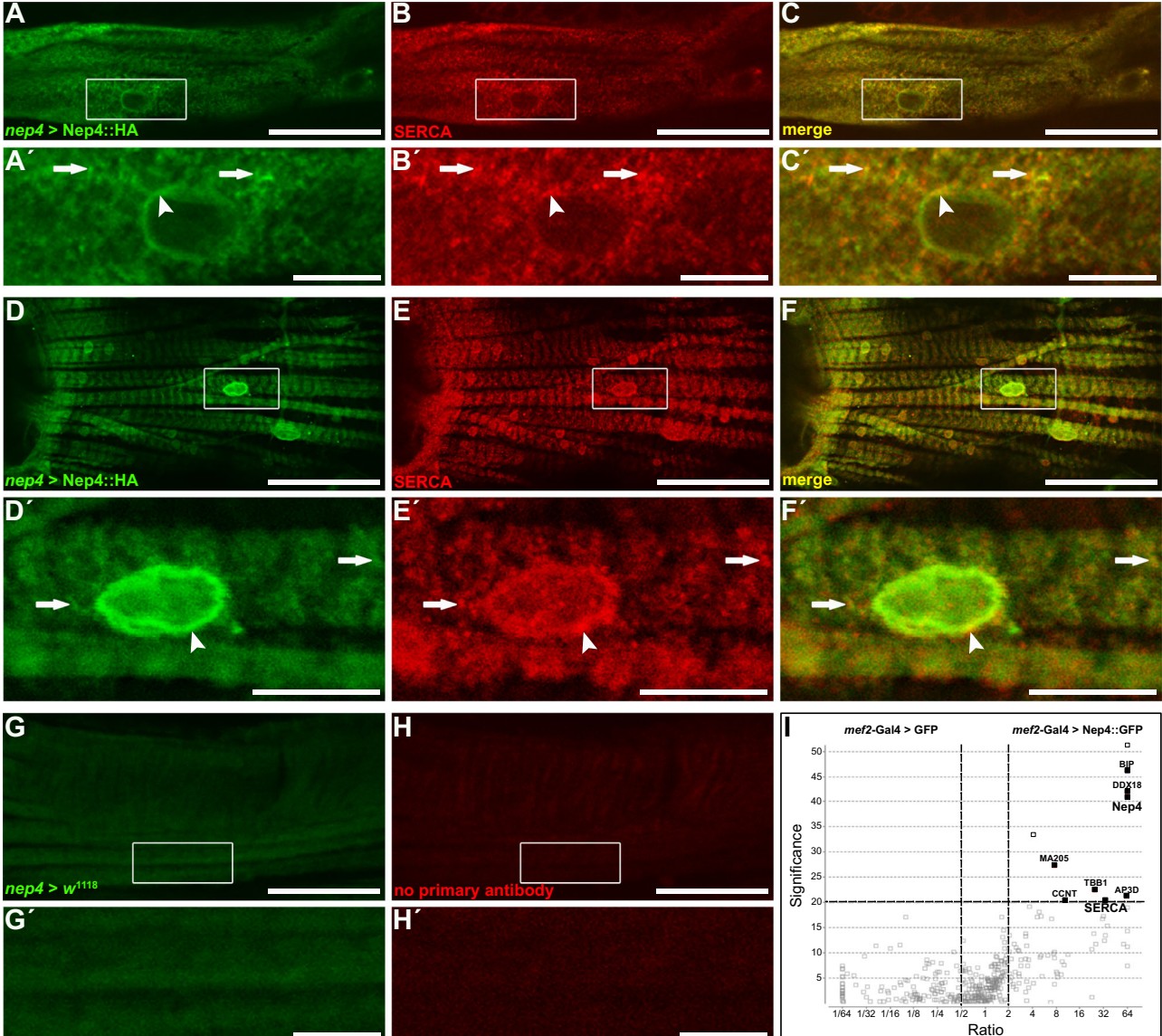

**Fig. 3 | Nep4 partially colocalizes with SERCA in heart tissue. A, D** Nep4::HA was expressed under the control of the native *nep4* enhancer and labeled with a monospecific anti-HA antibody (*nep4* > Nep4::HA). **B, E** SERCA was labeled with a monospecific antibody detecting the endogenous protein (SERCA). Optical slices of adult heart muscle fibers (**A**−**C**) or ventral longitudinal muscle fibers (**D**−**F**) are shown. Scale bars: 50 µm; ventral view, anterior left. Boxes indicate areas of higher magnification, as depicted in (**A´**−**C´**) and (**D´**−**F´**). Scale bars: 10 µm. Nep4::HA colocalizes with SERCA in membranes contiguous with the nuclear membrane (**C´**, **F´**, arrowheads). In addition, both proteins partially colocalize in a punctate

manner along the muscle fibers (**C´**, **F´**, arrows). Control stainings lacking the UAS-Nep4::HA construct (*nep4* > $w^{1118}$, **G, G´**) or primary SERCA antibodies (**H, H´**) do not exhibit any signal above background. **I** Volcano plot depicting the results of pull-down assays using Nep4::GFP as bait and free cytoplasmic GFP as control. Bait proteins were expressed in third instar larval muscle tissue. SERCA coprecipitates with Nep4::GFP. Data are based on three individual biological replicates. Open outlined squares depict proteins with quantification being based on only one detected peptide. Corresponding candidates were excluded from further analysis. A significance value of 20 corresponds to $p < 0.01$ (one-way ANOVA).

Considering the functional conservation of SERCA-regulatory peptides in metazoa[35,36], we investigated whether the *Drosophila* SCL peptides are also cleaved by human neprilysin (NEP). Interestingly, the human enzyme hydrolyzed the *Drosophila* peptides with the same specificity as Nep4, yielding largely identical cleavage products (Fig. 5C, D). Furthermore, SLN, the vertebrate ortholog of SCL, was also hydrolyzed by both enzymes in a similar manner (Fig. 5E, F). In this regard, the SR-luminal domain of human SLN (WLLVRSYQY, $SLN_{lum}$) was analyzed for NEP- as well as Nep4-catalyzed hydrolysis. The peptide was selected because of two reasons: (i) it represents a potent inhibitor of SERCA activity and (ii) it exhibits structural and functional homology to *Drosophila* SCL, including an intraluminal C-terminus of considerable size and mechanistic relevance[35,37,38]. Both human NEP and *Drosophila* Nep4 cleaved $SLN_{lum}$ predominantly

between Leu-2 and Leu-3 as well as between Ser-6 and Tyr-7 (Fig. 5E, F), which confirms SLN as a NEP substrate in vitro and produces initial evidence that homeostasis and turnover of SERCA-regulatory micropeptides are controlled via similar mechanisms in humans and flies. All analyzed peptides, along with the individual Nep4- or NEP-specific hydrolysis positions, are depicted in Supplementary Table 1.

**Neprilysin-mediated hydrolysis impairs membrane anchoring of SERCA-inhibitory micropeptides in S2 cells**

To investigate the physiological significance of Nep4/NEP-mediated SCL/SLN hydrolysis, we analyzed the consequences of the cleavage events at the molecular level. Therefore, CLIP-tagged full-length SCLA, SCLB, and SLN were expressed in *Drosophila* S2 cells, either with or without co-expression of Nep4 (SCLA, SCLB) or human NEP (SLN)

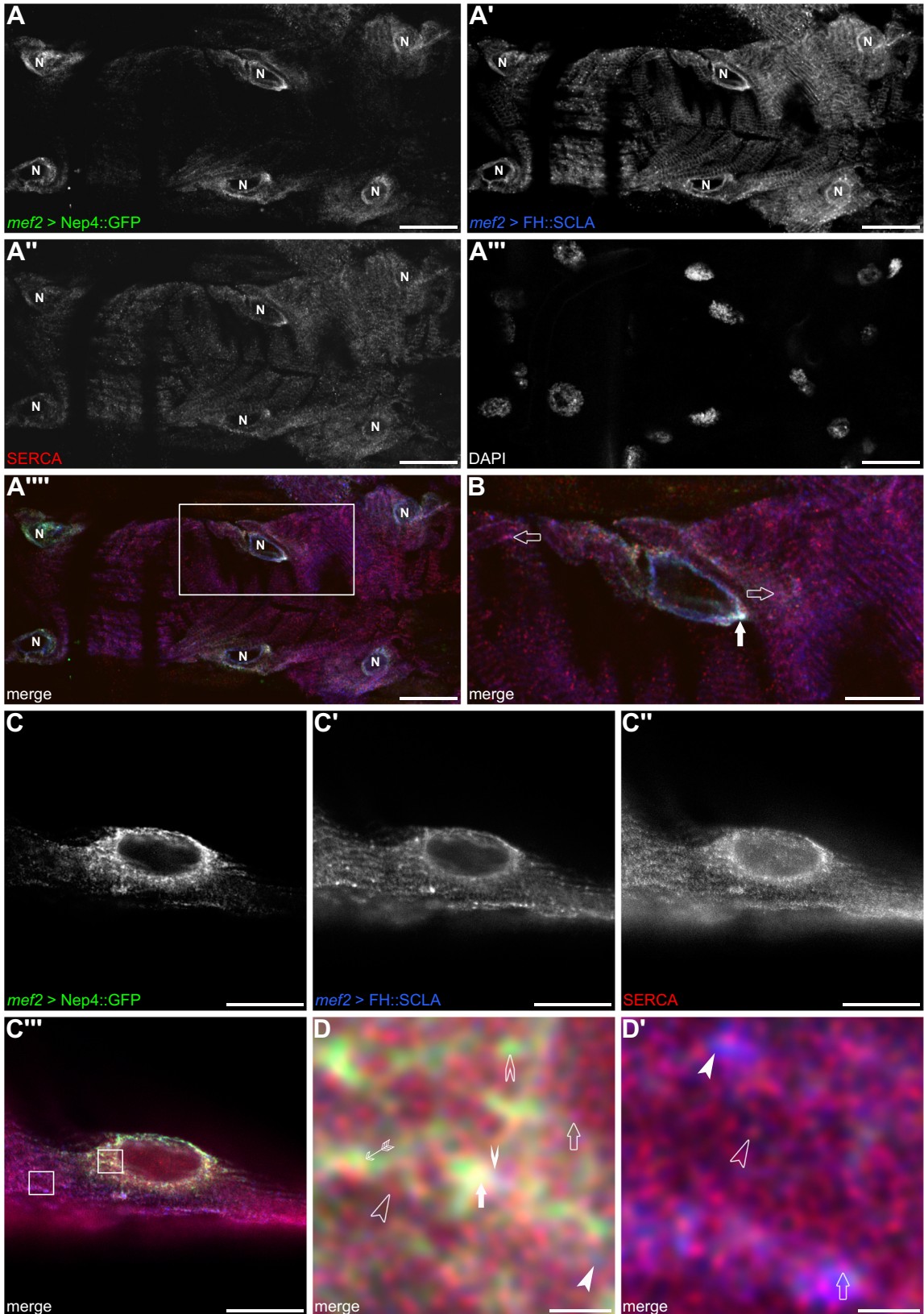

(Fig. 6A–C). Consistent with the localization pattern in heart and body wall muscles (Figs. 3 and 4 and Supplementary Figs. 1–3), the full-length peptides localized to the outer nuclear membrane and the ER of the cells (Fig. 6A, arrows), with only minor signals being present in nucleoplasm or cytoplasm. A similar localization pattern of full-length SCLA and SCLB in S2 cells has been reported previously[11]. Strikingly,

co-expression of Nep4 or NEP resulted in partial relocalization of the peptides to the cytoplasm, as indicated by reduced signal overlap with the ER marker Calnexin (Fig. 6A, arrowheads), suggesting a reduced membrane association of the cleaved peptides. To corroborate this indication, we analyzed cytoplasmic and membrane-enriched fractions generated from correspondingly transfected S2 cells by in-gel

**Fig. 4 | Nep4 partially colocalizes with Sarcolamban A.** GFP-tagged Nep4 (*mef2* > Nep4::GFP, **A**, **A´´´´**, **B**, **C**, **C´´´**, **D**, **D´**) and FH-tagged Sarcolamban A (*mef2* > FH::SCLA, **A´**, **A´´´´**, **B**, **C´**, **C´´´**, **D**, **D´**) were expressed under the control of the muscle-specific *mef2* enhancer and labeled with monospecific antibodies against the GFP- or the FH-tag. SERCA was labeled with a monospecific antibody detecting the endogenous protein (SERCA, **A´´**, **A´´´´**, **B**, **C´´**, **C´´´**, **D**, **D´**). DAPI was used as a nuclear marker (**A´´´**). Optical slices of third instar larval heart tissue are shown. The box in **A´´´´** indicates an area of higher magnification as depicted in (**B**). (**A–A´´´´**, **B**) Nep4, SERCA, and SCLA signals colocalize predominantly around the nuclei (**B**, solid arrow). More distant from the nuclei, only SERCA and SCLA signals are visible (**B**, open arrows). Scale bars: 20 μm (**A–A´´´´**); 10 μm (**B**); ventral view, anterior left. (**C–D´**) STED images of analogously stained cardiac tissue. Boxes in **C´´´** indicate areas of higher magnification as depicted in (**D**, **D´**). Overlap between Nep4, SERCA, and SCLA signals is visible predominantly around the nucleus (**D**, solid arrow). Individual colocalization between SCLA and SERCA (**D**, open arrow), Nep4 and SERCA (**D**, feathered arrow), or Nep4 and SCLA (**D**, solid winged arrowhead) occurs occasionally. Individual signals of SERCA (**D**, open arrowhead), Nep4 (**D**, open winged arrowhead), and SCLA (**D**, solid arrowhead) are present as well. More distant from the nucleus, only SERCA and SCLA are detected, while Nep4 is largely absent (**D´**). For SERCA and SCLA, again a colocalizing portion (**D´**, open arrow) as well as individual SERCA (**D´**, open arrowhead) and SCLA signals (**D´**, solid arrowhead) are present. Scale bars: 10 μm (**C-C´´´**); 1 μm (**D**, **D´**).

fluorescence detection and correlated Western blots. In absence of Nep4 or NEP, all peptides were present predominantly in the membrane-enriched fraction, but shifted into the cytoplasmic fraction in response to added expression of the individual peptidases (Fig. 6B). This effect was observed for all three peptides tested, yet the extent of relocalization was peptide-specific. While in the case of SCLA, the ratio between membrane-enriched and soluble fraction shifted from 1.46 (-Nep4) to 0.65 (+Nep4, factor 2.3), the respective ratio for SCLB changed from 2.92 (-Nep4) to 1.39 (+Nep4, factor 2.1). SLN exhibited the strongest response to neprilysin expression, with a shift in ratio from 12.31 (-NEP) to 3.04 (+NEP, factor 4.1). The free CLIP-tag was considered a fully soluble control (ratio: 0.17, Fig. 6C). In addition to expressing the full-length peptides together with the individual neprilysins, we also analyzed truncated forms of SCLA, SCLB, and SLN. In this regard, we expressed the predominant Nep4- or NEP-hydrolysis products identified above (Fig. 5) and evaluated their respective localization in relation to the corresponding full-length peptides. All truncated peptides exhibited a considerably reduced membrane association (Supplementary Fig. 6), with the extent being similar to the effects observed as a result of co-expressing the individual neprilysins along with the full-length peptides (Fig. 6). These results indicate that Nep4-mediated SCL cleavage and NEP-mediated SLN cleavage impair membrane localization of the respective peptides in a similar manner.

**Neprilysin-mediated hydrolysis affects the amount and oligo-merization states of SERCA-inhibitory micropeptides in muscle tissue**

In muscle cells of transgenic animals, reduced membrane localization in response to Nep4 overexpression was not observed for the *Drosophila* peptides (Fig. 6D and Supplementary Fig. 7). However, in this tissue we detected species of higher molecular weight for both SCLA and SCLB that occurred in addition to the monomeric peptides (Fig. 6D). Corresponding constructs were stable in SDS-PAGE, thus probably representing SDS-resistant oligomers. Based on the apparent molecular masses (monomers: 17 kDa, oligomers: 68 kDa), tetrameric forms of the peptides are most likely. Of note, a similar resistance to denaturing conditions has also been confirmed for vertebrate PLN oligomers, with a pentameric form being predominant[39,40]. To our knowledge, this is the first indication of the ability of SCL peptides to oligomerize. Strikingly, for both peptides the high molecular weight species were largely absent in animals that overexpressed Nep4 in a muscle-specific manner (Fig. 6D, E), thus confirming in vivo relevance of the Nep4-mediated SCLA/SCLB cleavage and indicating that Nep4-mediated hydrolysis impairs the ability of both peptides to oligomer-ize. For SCLA, increased Nep4 expression also reduced the overall amount of the peptide in muscle cells by 56.9%, relative to animals with endogenous Nep4 expression, while overall SCLB levels were not significantly affected (Fig. 6E). The latter result was based on a tendency of monomeric SCLB to accumulate in animals with elevated Nep4 expression, thus compensating for the significant reduction in the amount of oligomeric SCLB and indicating higher stability of Nep4-hydrolyzed SCLB, compared to hydrolyzed SCLA, in muscle tissue (Fig. 6D, E).

To further substantiate the indication that neprilysin-mediated SCL / SLN hydrolysis represents an evolutionarily conserved mechanism to regulate the abundance and localization of the peptides, and thus SERCA activity, we analyzed the subcellular localization pattern of NEP and SERCA in human ventricular cardiomyocytes (Supplementary Fig. 8). Both enzymes exhibited a highly similar localization, with the main signals being concentrated along the Z-discs of the muscle fibers (Supplementary Fig. 8A´´, arrow) and in membranes continuous with the nuclear membrane (Supplementary Fig. 8A´´, arrowhead). In control stainings lacking primary antibodies, no signal above background was observed (Supplementary Fig. 8A´´´). These data indicated colocalization of NEP and SERCA in SR membranes of human cardiomyocytes, thus supporting the notion that neprilysin-mediated regulation of SERCA activity is relevant in humans and that the underlying molecular mechanisms may be comparable in flies and humans.

## Discussion

While the principal mechanisms of muscle contraction in metazoa have been intensively studied, adaptive responses to variable physiological requirements are not entirely understood at the molecular level. However, SERCA is a major player involved in adaptation[3,4,10,36]. As known for vertebrates[10], and more recently confirmed in insects[11], the activity of SERCA is controlled by SR membrane integral micropeptides that bind to the enzyme and modulate its activity. Well-characterized vertebrate peptides include PLN, SLN, MLN, and DWORF[12,13,20–22], while SCL represents a *Drosophila* ortholog of the SERCA-regulatory micropeptides[11]. Based on the critical physiological relevance, vertebrate and fly loss-of-function mutants for these peptides exhibit compromised Ca²⁺ transients in heart cells concomitant with heart arrhythmia. In addition to the muscle-specific factors, endoregulin (ELN) and another-regulin (ALN) have been identified as SERCA-inhibiting micropeptides in non-muscle tissues, indicating a conserved mechanism for the control of intracellular Ca²⁺ dynamics in both muscle and non-muscle cell types[35]. To date, no mechanism has been identified that directly regulates the turnover of any of the micropeptides within the SR membrane.

This study demonstrates that modulating the expression of the endopeptidase Nep4 significantly affects SERCA activity in *Drosophila*, thus establishing a correlation between neprilysin activity and the regulation of SERCA. A high physiological relevance is confirmed by the fact that altering *nep4* expression phenocopies characteristic effects of SCL knockout, including abnormal heart rhythmicity as well as impaired SERCA activity and SR Ca²⁺ load (Figs. 1 and 2). By confirming Nep4-mediated hydrolysis of SERCA-regulatory SCL peptides (Fig. 5 and Supplementary Fig. 5), we provide evidence that neprilysin-mediated cleavage of these peptides is the mechanistic basis of the described phenotypes. The finding that the catalytically-active enzyme, and not the inactive variant, affected heart function and SERCA activity (Figs. 1 and 2), supports this premise as it confirms aberrant enzymatic activity, and thus abnormal peptide hydrolysis, as a causative parameter.

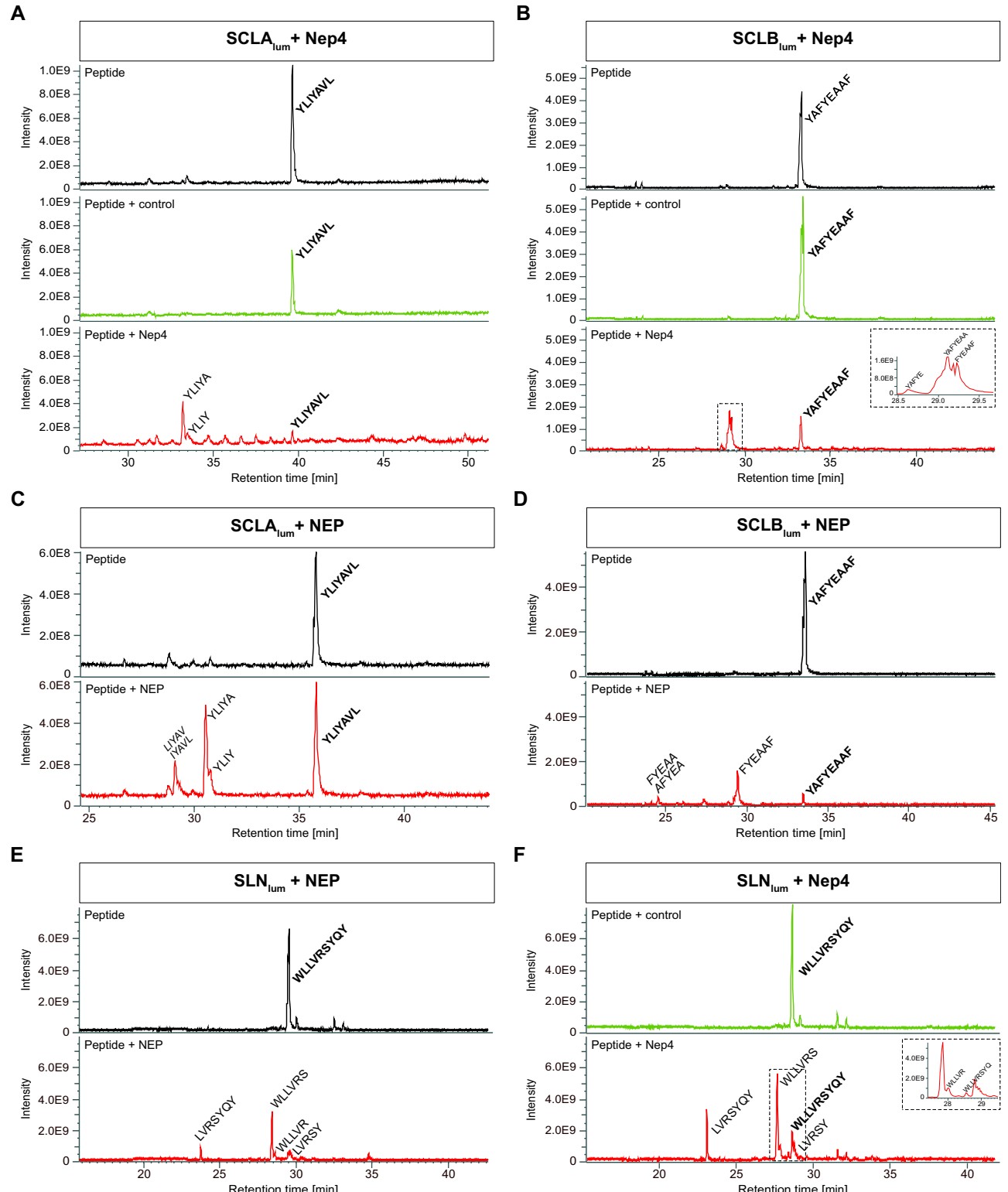

Interestingly, we found that in S2 cells both, the *Drosophila* SCL peptides as well as human SLN exhibited a significantly reduced membrane anchoring ability as a result of co-expression of Nep4 or NEP, respectively (Fig. 6A–C). A similar effect was observed if the Nep4/NEP hydrolysis products, identified in the in vitro cleavage assays (Fig. 5), were expressed in S2 cells. Again, Nep4-cleaved SCL as well as human NEP-cleaved SLN exhibited a significantly reduced membrane localization, relative to the non-cleaved full-length peptides (Supplementary Fig. 6). A milder relocalization effect was

observed for truncated SCLA and SCLB, and a stronger was present with truncated SLN (Fig. 6 and Supplementary Fig. 6). Given the distinct cleavage products (Fig. 5, Supplementary Fig. 6), the larger truncation of SLN (7 amino acids) in comparison to SCLA (2 amino acids) and SCLB (6 amino acids) may to some extent account for the individual effects. This possibility is supported by data confirming a similar degree of relocalization for human SLN and PLN in response to consecutive C-terminal truncation[37,41], thus emphasizing the functional relevance of the corresponding C-termini.

**Fig. 5 | Neprilysins hydrolyze the luminal domain of SERCA-inhibitory micro-peptides.** Depicted are total ion chromatograms of the luminal part of *Drosophila* Sarcolamban A (SCLA$_{lum}$, **A**, **C**), *Drosophila* Sarcolamban B (SCLB$_{lum}$, **B**, **D**), and human sarcolipin (SLN$_{lum}$, **E**, **F**). Full-length peptides (bold) are detected under all applied experimental conditions (peptide only, black chromatograms; peptide incubated with control preparation, green chromatograms; peptide incubated with purified enzyme, red chromatograms). Specific cleavage fragments are detected only after the addition of enzyme. **A** Incubation of SCLA$_{lum}$ (**YLIYAVL**) with *Drosophila* Nep4 results in the formation of YLIYA and YLIY fragments (red chromatogram). **C** The same fragments are generated by human neprilysin (NEP)-mediated hydrolysis of SCLA$_{lum}$. An additional peak corresponds to either *LIYAV* or *IYAVL*. **B** Incubation of SCLB$_{lum}$ (**YAFYEAAF**) with *Drosophila* Nep4 results in the

formation of FYEAAF, YAFYEAA, and YAFYE fragments (red chromatogram). **D** Human NEP-mediated hydrolysis of SCLB$_{lum}$ generates predominantly FYEAAF. In addition, minor amounts of *FYEAA* or *AFYEA* are produced. **E** Incubation of SLN$_{lum}$ (**WLLVRSYQY**) with human NEP results in the formation of WLLVRS, LVRSYQY, LVRSY, and WLLVR fragments (red chromatogram). **F** Incubation of SLN$_{lum}$ (**WLLVRSYQY**) with *Drosophila* Nep4 results in the formation of WLLVRS, LVRSYQY, and LVRSY fragments (red chromatogram). Insets depict areas of magnification indicated by the dashed boxes. Italicized fragments could not be assigned to one exclusive peptide sequence. Y-axes show absolute peak intensities, X-axes depict retention times. Individual cleavage assays were repeated at least three times.

In muscle tissue of transgenic animals, altered membrane localization in response to Nep4 overexpression was not observed for SCLA or SCLB. This was mainly due to the fact that, unlike the results in S2 cells, the peptide amounts detectable in the cytoplasmic fractions prepared from transgenic animals were very low, even if Nep4 was overexpressed. As a consequence, quantifying the ratio between soluble and membrane-bound peptides was rather error-prone (Fig. 6D and Supplementary Fig. 7). One explanation for this outcome could be an efficient degradation of the soluble Nep4-cleaved peptides in the cytoplasm of muscle cells, which may not occur that efficiently in S2 cells. Of note, instability and rapid intracellular degradation have already been reported for truncated forms of vertebrate PLN[16] and SLN[37].

However, a high in vivo relevance of the Nep4-mediated SCL cleavage was confirmed by the fact that muscle-specific over-expression of Nep4 significantly reduced the overall amount of SCLA in this tissue and, furthermore, largely abolished the formation of high molecular weight species of SCLA and SCLB, presumably representing peptide oligomers (Fig. 6D, E). While the reduced SCLA levels clearly indicate that Nep4 is required to control the amount of this peptide within the SR membrane, the reduced oligomerization observed for both SCLA and SCLB is equally remarkable, considering that oligomerization is also a well-known and physiologically highly relevant characteristic of vertebrate PLN[10]. Here, oligomerization represents a mechanism for the storage of active PLN monomers, playing a key role in SERCA regulation[42,43]. Given our observation that SCL may also form oligomers (Fig. 6D, E), a similar mechanism could be present in *Drosophila*, with the Nep4 cleavage activity controlling both, the overall amount of the peptides, as well as the ratio between monomeric and oligomeric SCL. Of note, SCLA and SCLB oligomers were detected by SDS-PAGE, indicating high thermal stability and detergent resistant association. A similar resistance to denaturing conditions has also been reported for vertebrate PLN oligomers[39,40].

Together with the data from S2 cells, these results indicate that (i) Nep4 hydrolyzes SCLA and SCLB in vivo, (ii) this cleavage event reduces membrane localization of the peptides as well as their ability to oligomerize, and (iii) cleavage considerably reduces the overall amount of SCLA in muscles. In this regard, S2 cells may feature the principal molecular mechanism, while in muscle tissue, the highly adapted conditions result in a more complex regulation. Here, especially the high abundance of SERCA may be crucial, which has been shown to affect the oligomeric structure of vertebrate PLN, with an increasing SERCA/PLN ratio causing depolymerization of pentameric PLN[44]. Consistent with our results on the *Drosophila* SCL peptides, analyses on human PLN found (i) a reduced membrane localization, (ii) a reduced ability of the peptide to form oligomers, and (iii) a reduced binding affinity to SERCA as a result of consecutive C-terminal truncation[41]. Thus, in addition to emphasizing the physiological relevance of the corresponding C-termini, these results corroborate the indication that the amount and function of *Drosophila* and vertebrate SERCA-inhibitory micropeptides are regulated in a mechanistically similar manner.

Because interaction between the regulatory micropeptides and SERCA occurs within the SR membrane[10,11,13], it appears likely that loss of membrane localization represents the critical physiological event inactivating the peptides. This inactivation is presumably required to prevent the peptides from accumulating within the SR membrane and, eventually, from abnormally binding and inhibiting SERCA (Fig. 7). The fact that C-terminal truncation likewise affects membrane anchoring of human PLN[41], human SLN (ref. 37, Fig. 6 and Supplementary Fig. 6), and *Drosophila* SCL (Fig. 6 and Supplementary Fig. 6) indicates the presence of an evolutionarily conserved mechanism to control membrane association and general abundance of corresponding peptides.

Significantly, up to now no candidate enzyme that adequately catalyzes C-terminal cleavage in vivo has been identified in any organism. Our data indicate that Nep4 represents a corresponding factor in *Drosophila* and that the peptidase is necessary and sufficient to control the amount of SERCA-regulatory peptides within the SR membrane. Under *nep4* overexpression conditions, SCL peptides are excessively cleaved, while *nep4* knockdown results in peptide accumulation. As a result of both interventions, appropriate regulation of SERCA activity is impaired (Fig. 2), which eventually causes the observed phenotypes (Figs. 1, 2, and 7).

In contrast to vertebrates that express multiple SERCA genes in a time- and tissue-specific manner[4,45], in *Drosophila* only one corresponding gene is known (CaP60A[46]). Interestingly, this reduced complexity also applies to the SERCA-regulatory micropeptides. Currently, the only identified SERCA-regulatory micropeptides in *Drosophila* are SCLA and SCLB, with expression being confirmed in both cardiac and somatic muscle tissue[11]. By contrast, vertebrate genomes contain at least four corresponding genes, each of them exhibiting specific expression patterns. While PLN and SLN are expressed in cardiac and slow skeletal muscle, MLN appears to be specific to all skeletal muscles[12,13,20] and DWORF, the only SERCA-activating micropeptide identified to date, specific to the heart and soleus[21]. This reciprocal expansion of SERCA and regulatory peptide family members during evolution suggests that the two families have coevolved as an effective and general mechanism to control Ca$^{2+}$ handling in muscle cells. In this respect, the situation in *Drosophila*, with only one SERCA and two micropeptide genes being present, would represent the phylogenetically ancient status, while a similar, yet more specialized regulation occurs in vertebrates. Our finding that altering *nep4* expression has similar effects in heart and body wall muscle cells supports this idea by indicating similar underlying physiologies in both contractile tissues (Fig. 1). Moreover, considering our data on human NEP (Figs. 5 and 6 and Supplementary Figs. 6 and 8), it appears likely that also the neprilysin-mediated regulation of micropeptide abundance is evolutionarily conserved. We could show that both human NEP and *Drosophila* Nep4 hydrolyze the luminal domain of human SLN with identical cleavage specificity (Fig. 5E, F). Furthermore, the effects on membrane anchoring were similar for truncated SCL and truncated SLN (Fig. 6A–C and Supplementary Fig. 6), and also the localization pattern of NEP and SERCA in human ventricular cardiomyocytes (Supplementary Fig. 8) was reminiscent of the corresponding pattern in the *Drosophila* heart

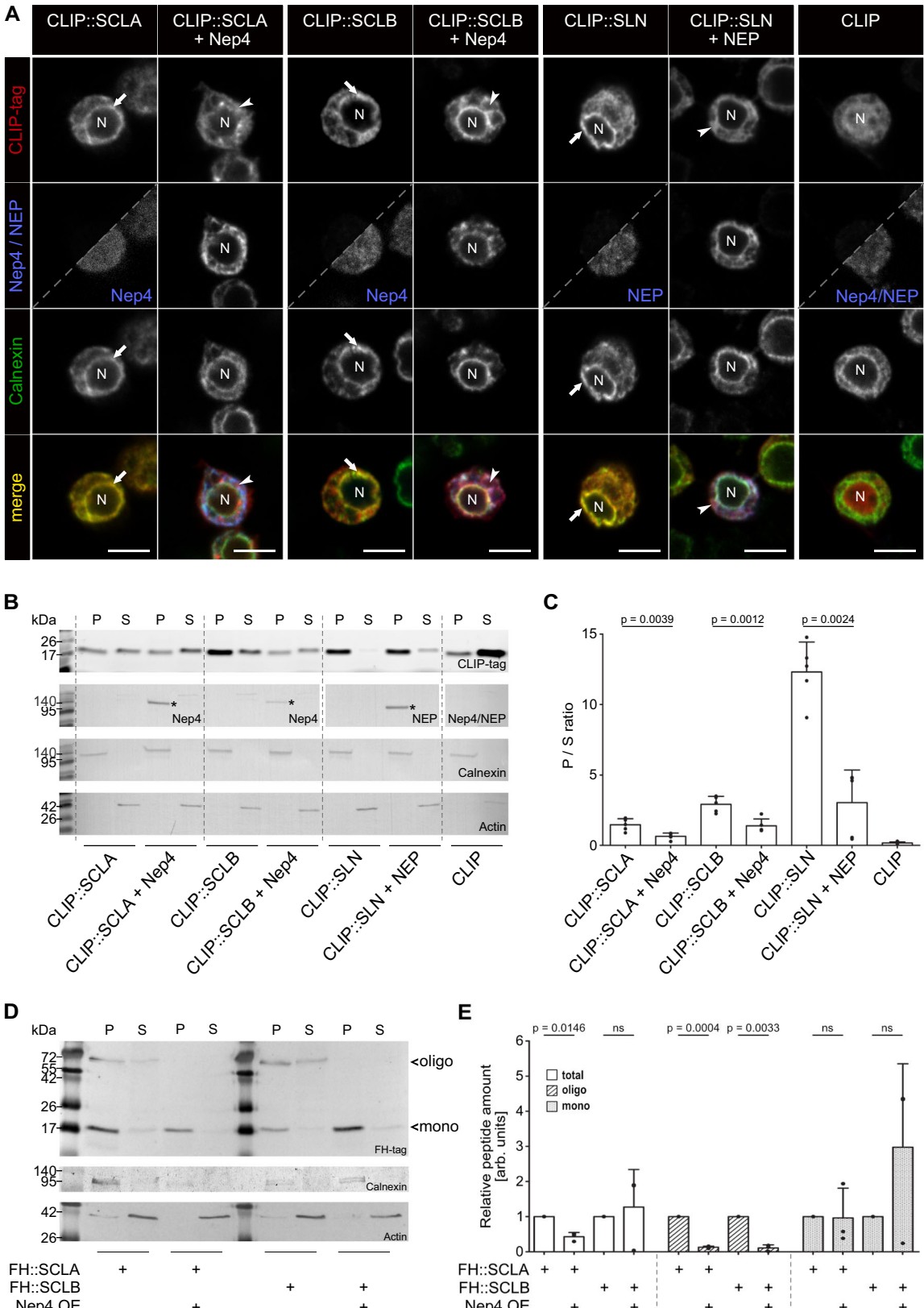

(Figs. 3 and 4 and Supplementary Figs. 1 and 2). Interestingly, NEP expression has been reported to be elevated in cardiomyocytes of patients suffering from aortic valve stenosis or dilated cardiomyopathy[47]. In that study, it was speculated that elevated NEP activity in these patient groups resulted in increased degradation of local bradykinin and ANP/BNP, and thus reduced protection from the

progression of hypertrophy and fibrosis. However, in view of our data, an alternative scenario may be more likely. We found that in human ventricular cardiomyocytes NEP localized in direct proximity to SERCA, which indicates colocalization in membranes of the SR (Supplementary Fig. 8). Therefore, it is conceivable that the increased NEP expression results in increased cleavage of SLN and correspondingly

**Fig. 6 | Neprilysin-hydrolyzed SERCA-inhibitory micropeptides exhibit reduced membrane anchoring. A** All constructs were expressed in *Drosophila* S2 cells as full-length peptides (CLIP::SCLA; CLIP::SCLB; CLIP::SLN), either with or without co-expression of Nep4 (SCLA, SCLB) or human NEP (SLN). CLIP-tag and Nep4 or NEP were visualized by immunostainings as indicated. Split images are overexposed in the lower right part to confirm the presence of cells. Anti-Calnexin antibodies were applied as ER membrane marker. Cells expressing the free CLIP-tag were used as a soluble control. While full-length SCL/SLN peptides mainly localize to the ER (arrows), co-expression of Nep4 or NEP results in reduced signal overlap of the peptides with the ER marker (arrowheads). Scale bars: 5 μM. **B** Subcellular fractions of *Drosophila* S2 cells expressing the indicated CLIP-tagged SCL or SLN constructs, with or without co-expression of Nep4 or NEP. Western blot analysis was performed with anti-Nep4 and anti-NEP antibodies, as indicated, to confirm peptidase expression (asterisks), and with anti-Calnexin antibodies (a marker for ER membranes) and anti-Actin antibodies (cytosolic marker) to confirm the identity of the individual fractions. P = pellet (membrane-enriched); S = supernatant. **C** Peptide-

specific ratios between membrane-enriched (P) and soluble (S) fractions were determined by pixel intensity measurements. The diagram depicts the resultant mean values (+ SD) of five individual biological replicates. Significant differences between the individual peptide-specific ratios are indicated (paired t-test, two-tailed). **D** Subcellular fractions of *Drosophila* third instar larvae expressing the indicated SCL constructs in a muscle-specific manner (*mef2*-Gal4) were analyzed by Western blot. Peptides were detected with anti-HA antibodies. Calnexin or Actin signals were used to confirm the identity of the individual fractions. Actin signals were used for normalization. P = pellet (membrane-enriched); S = supernatant; mono = peptide monomer; oligo = peptide oligomer. **E** Relative amounts of peptide oligomers (oligo), peptide monomers (mono), and the sum of both (total) were determined by pixel intensity measurements. For each of the indicated SCL/Nep4 combinations, the combined signals from pellet and supernatant fractions, as depicted in **D**, were evaluated. Resultant data represent mean values (+SD) of three individual biological replicates (paired t-test, two-tailed). Source data are provided as a Source data file.

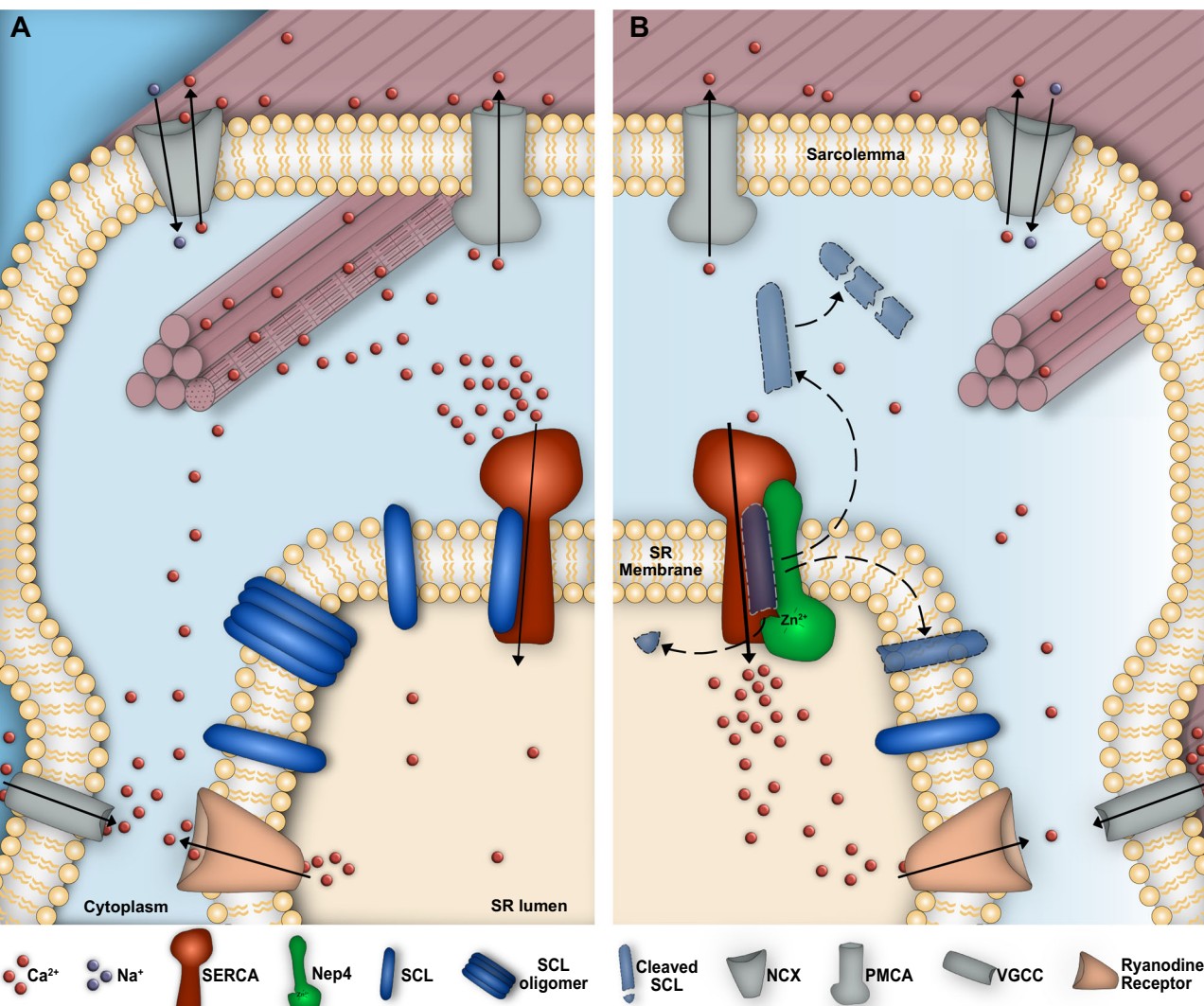

**Fig. 7 | Neprilysin-mediated hydrolysis of SERCA-inhibitory micropeptides is essential to control SERCA activity. A** Without Nep4, Sarcolamban (SCL) peptides occur in a monomeric and an oligomeric state and accumulate within the sarcoplasmic reticulum (SR) membrane, resulting in abnormal binding and inhibition of SERCA and a consequential reduction in the SERCA-mediated Ca²⁺ transport.

**B** Nep4-mediated SCL hydrolysis reduces the ability of the peptides to oligomerize and releases them from the SR membrane, thus preventing SCL accumulation and excessive SERCA inhibition. Released peptides become degraded in the cytoplasm. VGCC voltage-gated L-type Ca²⁺ channel, RyR Ryanodine receptor, NCX Na⁺/Ca²⁺ exchanger, PMCA plasma membrane Ca²⁺ ATPase.

enhanced SERCA activity and cardiomyocyte contractility. Thus, elevated NEP expression in aortic valve stenosis or dilated cardiomyopathy patients may represent a physiological response to restore cardiac ejection fraction and to ameliorate the effects of cardiac disease, rather than being causative.

While further research is necessary to assess these issues in detail, the present study introduces neprilysin activity as an efficient regulator of SERCA. As SERCA is an important therapeutic target in myocardial disorders, our data can represent a valuable basis for the development of innovative therapies against predominant heart and muscle diseases.

## Methods

### Ethics

The use of human tissue from explanted hearts for research purposes was according to the convention of Helsinki and accepted by the local ethics committee (ethics commission of the faculty of the Ruhr University Bochum, located in Bad Oeynhausen, vote 21/2013). Informed consent was obtained from all participants.

### Fly strains

The following *Drosophila* lines were used in this work. $w^{1118}$ (RRID:BDSC_5905) was used as a control strain. RRID:BDSC_59041 that expresses GFP fused to an ER / SR targeting signal from Cytochrome b5 under the control of UAS was applied as a control in pull-down assays. Driver lines were *mef2*-Gal4 (RRID:BDSC_27390), *tinC*-Gal4 (R. Bodmer, Sanford Burnham Medical Research Institute, San Diego, CA, USA), and *nep4*-Gal4[25,29]. UAS lines were UAS-Nep4A[25], UAS-Nep4A$_{inact}$ (catalytically inactive form (E873Q)[25]), UAS-FH::SCLA[11], and UAS-FH::SCLB[11]. UAS-roGFP and UAS-Nep4::roGFP lines were established by cloning the *roGFP2*[33] or *nep4::roGFP2* coding sequence into the pUAST vector[48]. The latter construct included a short in-frame spacer sequence (gcgggcgga) between the *nep4A* and *roGFP2* coding sequences. The resulting constructs were subjected to P-element-based transformation using commercial services (Best Gene Inc., CA, USA). Flies expressing GCaMP3 in a cardiac-specific manner (UAS-GCaMP3/UAS-GCaMP3; *tinCΔ4*-Gal4, UAS-GCaMP3/*tinCΔ4*-Gal4, UAS-GCaMP3[49]) were applied to measure $Ca^{2+}$-transients in cardiomyocytes. Knockdown of *nep4* was achieved using line 100189 (KK library, Vienna *Drosophila* Resource Center [VDRC]). High *nep4*-specific knockdown efficiency of the respective hairpin was confirmed previously[24,25]. A second *nep4*-specific RNAi construct (line 16669, GD library, VDRC) did not significantly reduce *nep4* transcript levels[25]. It was therefore excluded from further analysis.

### Generation of expression constructs

For heterologous expression of Nep4B in *Sf*21 cells, the *nep4B* coding sequence was fused to a C-terminal His-tag by appropriate primer design and cloned downstream of the polyhedrin promoter into a pFastBac Dual vector (Thermo Fisher Scientific, Waltham, MA, USA). To track transfection efficiency, an *eGFP* reporter gene was inserted into the same vector under the control of the p10 promoter. Generation of recombinant bacmids and transfection/infection was performed according to the Bac-to-Bac baculovirus expression system manual (Life Technologies, Carlsbad, CA, USA).

Plasmids for expression of CLIP-tagged peptides, human NEP or *Drosophila* Nep4 in S2 cells were generated based on an *E. coli/S. cerevisiae/D. melanogaster* triple-shuttle derivative of the pAc5.1/V5-His vector (Thermo Fisher Scientific) adapted for cloning by homologous recombination in vivo. The respective vector was constructed by in vivo recombination in yeast, inserting *URA3* and a 2 μm sequence for selection and propagation, respectively[50]. Constructs expressing truncated peptides were generated using the Q5 Site-Directed Mutagenesis Kit (New England Biolabs, Ipswich, MA, USA) according to the manufacturer´s instructions.

### Motility and contraction analysis

Nep4 overexpression and knockdown were driven by *mef2*-Gal4. The driver line crossed to $w^{1118}$ was used as a control. Movement and contraction assays were conducted according to ref. 51. Here, wandering third instar larvae of each respective genotype were transferred onto glass petri dishes supplemented with millimeter paper. All movements were recorded with a standard video camera (Canon UC X10Hi, Canon, Tokyo, Japan). Movement speed was determined by calculating the distance covered in a continuous run of 10 s. The same run was used to count larval body contractions. Because increased Nep4 levels in muscle tissue affect body size[24], distance and speed were calculated relative to animal size.

### Analysis of cardiac function

For SOHA analysis, 1-week-old male flies were tranquilized on ice for 1 min, placed in a 60 mm petri dish containing a thin layer of Vaseline, and fixed by their spread wings with the ventral side up. Animals were then covered with artificial hemolymph solution containing 5 mM KCl, 8 mM MgCl$_2$, 2 mM CaCl$_2$, 108 mM NaCl, 1 mM NaH$_2$PO$_4$, 5 mM HEPES, 4 mM NaHCO$_3$, 10 mM trehalose, and 10 mM sucrose, pH 7.1[52]. For semi-intact heart preparations, the head as well as the ventral half of the thorax and abdomen and all abdominal internal organs, except for the dorsal vessel, were removed. After dissection, the preparation was allowed to rest for at least 10 min. Heartbeat recordings were done with a high-speed video camera (Basler piA-640) that was mounted onto an upright microscope (Leica DMLB) equipped with a 10× Leica Fluotar. Movies were captured with the software FireCapture (version 1.2, Torsten Edelmann) and processed with ImageJ (version 1.52n, National Institutes of Health, MD, USA). Heart parameters were analyzed using SOHA (semi-automated optical heartbeat analysis), a MATLAB application introduced by[23] and applied for the analysis of cardioactive peptides in *Drosophila* in ref. 53. SOHA utilizes overall darkness changes of a video (Frame Brightness Algorithm), combined with pixel-by-pixel intensity changes (Changing Pixel Intensity Algorithm), to allow calculation of different heart parameters. Here, heart rate [beats per minute, bpm] and arrhythmicity index [1/s] were determined. The latter was calculated by dividing the standard deviation of the heart period by the median of the heart period[23].

Measurements of $Ca^{2+}$ transients were conducted according to refs. 54,55. In detail, 1-week-old male flies were anesthetized with carbon dioxide, placed in a 60 mm petri dish containing Vaseline, and fixed with the ventral side up. The head and thorax were cut off to exclude neuronal influence on cardiac activity. The middle ventral region of the abdomen was opened and the internal organs were removed. Subsequently, the preparation was submerged in an oxygenated artificial hemolymph solution. $Ca^{2+}$ transients were recorded using a Carl Zeiss LSM 410 confocal microscope (Zeiss, Jena, Germany). GCaMP3 was excited at 488 nm with an Ar laser and the fluorescence signal was detected using a 505–530 nm band-pass detection filter. Changes in fluorescence of the conical chamber, localized in the first abdominal segment, were scanned, with an increase in fluorescence, followed by a decrease in fluorescence, reflecting the transient elevation of cytosolic [$Ca^{2+}$] that precedes contraction. Recordings of $Ca^{2+}$ transients were conducted for 25 s. The resulting images (1024 × 1024 pixel) were analyzed with ImageJ (version 1.52n, National Institutes of Health, MD, USA) and LabChart software (version 7.0, AD Instruments, CO, USA).

Analyzed hearts were not paced and parameters were only compared under the premise that spontaneous heart rates of semi-intact preparations were not significantly different between the individual groups of flies. $Ca^{2+}$ transient amplitudes were calculated according to the formula $F_{max} - F_0/F_0$ and depicted in arbitrary units of fluorescence. $F_{max}$ represents the maximal fluorescence signal obtained

during the systole, and $F_0$ corresponds to the minimal fluorescence recorded during the diastolic period. This approach normalizes differences in indicator concentration between cells, thus providing a plausible method for comparing ratios between different samples/genotypes. Relaxation was measured by calculation of Tau constant (s), based on the exponential decay of $Ca^{2+}$ transients. For estimating the sarcoplasmic reticulum $Ca^{2+}$ load, a caffeine pulse (10 mM) was applied to the perfusion media. This compound triggers the release of $Ca^{2+}$ through the ryanodine receptor[30]. The amplitudes of the caffeine-induced $Ca^{2+}$ transient and the $Ca^{2+}$ transients recorded during the 5 s preceding the caffeine pulse (pre-caffeine) were measured and $Ca^{2+}$ load was expressed as the ratio between the caffeine-induced $Ca^{2+}$ transient amplitude divided by the average of the corresponding pre-caffeine $Ca^{2+}$ transient amplitudes. SERCA activity was calculated by subtracting the inverse of the rate constant of caffeine-induced $Ca^{2+}$ transient decay ($Tau_{Caff}$) from that of the pre-caffeine transients ($Tau_{Ca}$), ($1/Tau_{Ca} - 1/Tau_{Caff}$). In this regard, the rate constant of $[Ca^{2+}]_i$ decay during a caffeine-induced $Ca^{2+}$ transient largely reflects the function of NCX, while both NCX and SERCA contribute to pre-caffeine $[Ca^{2+}]_i$ decay[30,56–63].

## Cell culture, subcellular localization, and fractionation

S2 cells (RRID:CVCL_Z232) were grown in Schneider´s *Drosophila* medium (Pan Biotech, Aidenbach, Germany). Transfection was done in 6-well plates or 15 cm$^2$ flasks using TransFectin (BioRad, Hercules, CA, USA) according to the manufacturer's instructions. For co-transfection, total plasmid amounts were adjusted to a maximum of 3 μg. For immunostaining, cells were fixed in 4% formaldehyde in phosphate-buffered saline (PBS) for 20 min and rinsed three times with PBS. Subsequently, samples were treated with permeabilization/blocking solution (0.1% Triton X-100, 2% bovine serum albumin [BSA] in PBS) for 20 min and rinsed again three times with PBS. To exclude a possible cross-reaction of antibodies, cells were sequentially stained. In a first step, samples were incubated with anti-Calnexin antibodies (RRID:AB_2722011, 1:200) overnight at 4 °C. Cells were washed with PBS (3x, 15 min each), blocked with ROTI ImmunoBlock (Roth, Karlsruhe, Germany) for 45 min, and incubated with secondary antibodies (anti-mouse-A546, RRID:AB_2534071, 1:200) diluted in ROTI ImmunoBlock for 90 min. Cells were washed as described before and incubated with anti-CLIP antibodies (RRID:AB_2827567, 1:100) and anti-Nep4/anti-NEP antibodies (anti-Nep4, RRID:AB_2569115, 1:500, monospecificity confirmed in;[28] anti-NEP, human, ab256494, 1:100, Abcam). Following another washing step, cells were blocked again with ROTI ImmunoBlock and incubated with secondary antibodies (anti-rat-A488, RRID:AB_2534074, 1:200; anti-rabbit-A647, RRID:AB_2338084, 1:100). Finally, cells were washed again as described above and mounted in Fluoromount-G (Thermo Fisher Scientific). Confocal images were captured with an LSM 800 microscope (Zeiss). For fractionation, 5 ml of transfected cells were harvested (300 × *g*, 3 min, room temperature [RT]) and labeled with 1 μM CLIP-Cell TMR-Star or 10 μM CLIP-Cell 505 (New England Biolabs) in 100 μl PBS at 37 °C for 30 min. The excess substrate was washed out with PBS for 15 min. Afterward, cells were resuspended in 500 μl 0.5 M NaCl including protease inhibitor (Promega, Madison, WI, USA.) and lysed using a glass-teflon homogenizer. Homogenates were centrifuged to remove cell debris (500 × *g*, 5 min, 4 °C) and the resulting supernatant was subjected to ultracentrifugation (100,000 × *g*, 1 h, 4 °C). The soluble cytoplasmic fraction was collected and prepared for SDS-PAGE, while the pellet was resuspended in 250 μl 0.5 M NaCl, using a 1 ml syringe with a 25 gauge needle. Following another ultracentrifugation step (100,000 × *g*, 45 min, 4 °C), the membrane-enriched pellet was resuspended again in the same buffer and prepared for SDS-PAGE. In-gel detection of CLIP-tagged constructs was done with a ChemiDoc MP Imaging System (BioRad). For quantification, pixel intensities were measured using Image Lab Software, Version 6.0.1 (BioRad).

For fractionation of tissue samples, transgenic third instar larvae were frozen in liquid nitrogen and homogenized. Homogenates were resuspended in 120 μl 0.5 M NaCl including protease inhibitor and centrifuged (10,000 × *g*, 15 min, 4 °C). The resulting supernatant was processed similar to the supernatant of S2 cells, yet 60 μl was used for pellet resuspension after ultracentrifugation. Western blot-based pixel intensities were quantified with a ChemiDoc MP Imaging System (BioRad) in combination with Image Lab Software, Version 6.0.1 (BioRad).

## Enzymatic cleavage assay

Heterologous expression of Nep4B was performed in *Sf*21 cells (RRID:CVCL_0518). Transfected (TransFectin, BioRad) and non-transfected *Sf*21 cells were cultured in 75 cm$^2$ flasks for 72 h and harvested by centrifugation (300 × *g*, 5 min). Subsequently, cells were resuspended in a 5 ml binding buffer (50 mM $NaH_2PO_4$, pH 7.9, 300 mM NaCl) and lysed with a glass-teflon homogenizer. The resulting homogenates were centrifuged (10,000 × *g*, 10 min) and the supernatants were subjected to gravity-flow-based His-tag purification according to the manufacturer's instructions (Protino Ni-NTA agarose, Macherey-Nagel, Düren, Germany). To measure enzymatic activity, 3 μl of Nep4B-containing (10 ng/ml, purified from *nep4B* transfected cells) and non-containing (from untransfected control cells) preparations were supplemented with 7 μl (250 ng) of individual peptides. To measure the activity of human NEP (transcript variant 1, $D^{53}$–$W^{750}$, N-terminal His-tag fusion, Enzo Life Sciences, Farmingdale, NY, USA), 1 μl (50 ng) of the enzyme was supplemented with 9 μl (250 ng) of individual peptides. All dilutions were prepared in hydrolysis buffer (100 mM NaCl, 50 mM Tris, pH 7.0). After 5 h of incubation (35 °C), 1 μl of each respective preparation was analyzed via mass spectrometry (see below). For measurements of untreated peptides, respective samples were diluted in hydrolysis buffer without the addition of enzyme and processed as described above. Peptides were synthesized at JPT Peptide Technologies (Berlin, Germany) with more than 90% purity. Individual cleavage assays were repeated at least three times.

## Pull-down assay

30 third instar larvae of each control genotype (*mef2*-Gal4 > UAS-roGFP or *mef2*-Gal4 > UAS-GFP.ER), as well as an equal weight of *mef2*-Gal4 > UAS-Nep4A::roGFP third instar larvae, were frozen in liquid nitrogen and homogenized. Homogenates were resuspended in 300 μl lysis buffer (150 mM NaCl, 50 mM Tris, 1 mM MgCl, 0.2% n-dodecyl-β-ᴅ-maltopyranoside [DDM]) and centrifuged (10,000 × *g*, 15 min). The resulting supernatants were incubated for 30 min at 4 °C with 50 μl of μMACS anti-GFP MicroBeads (Miltenyi Biotec, Auburn, CA, USA). Beads were immobilized in calibrated μ-columns (Miltenyi Biotec), which were equilibrated beforehand with 200 μl lysis buffer. After two washing steps using 200 μl lysis buffer, bound proteins were reduced by incubation in 50 μl reducing solution (10 mM dithiothreitol [DTT], 100 mM $NH_4HCO_3$) for 5 min at RT and an additional 30 min at 38 °C, followed by two washing steps with 20 mM Tris-HCl (pH 7.5). Subsequently, bound proteins were alkylated by the addition of 50 μl alkylation solution (54 mM Iodacetamid, 100 mM $NH_4HCO_3$) and incubation for 15 min in the dark. The columns were then washed twice with 100 μl digestion buffer (50 mM $NH_4HCO_3$, 5% acetonitrile), sealed, and incubated with 25 μl trypsin solution (0.01 μg trypsin/lysC in digestion buffer) overnight. Eluates were centrifuged (10,000 × *g*, 10 min) and subjected to mass spectrometry analysis (5 μl per sample). At least three independent biological replicates were analyzed for each genotype.

## Mass spectrometry

Reversed-phase chromatography was performed using the UltiMate 3000 RSLCnano System (Thermo Fisher Scientific). Samples were loaded onto a trap column (Acclaim PepMap 100 C18, 5 μm,

0.1 mm × 20 mm, Thermo Fisher Scientific) and washed with loading buffer (0.1% TFA in $H_2O$) at a flow rate of 25 µl/min. The trap column was switched in line with a separation column (Acclaim PepMap 100 C18 2 µm, 0.075 mm×150 mm, Thermo Fisher Scientific). Subsequently, bound peptides were eluted by changing the mixture of buffer A (99% water, 1% acetonitrile, 0.1% formic acid) and buffer B (80% acetonitrile, 20% water, and 0.1% formic acid) from 100:0 to 20:80 within 60 min. The flow rate was kept constant at 0.3 µl/min. Eluted compounds were directly electrosprayed through an EASY-Spray ion source (Thermo Fisher Scientific) into a Q Exactive Plus Orbitrap mass spectrometer (Thermo Fisher Scientific). Eluates were analyzed by measuring the masses of the intact molecules as well as the masses of the fragments, which were generated by higher-energy collisional dissociation (HCD) of the corresponding parent ion. For peptide cleavage assays, extracted ion chromatograms corresponding to the synthesized peptides or their respective proteolytic fragments were analyzed using FreeStyle Software (v1.3, Thermo Fisher Scientific). For analysis of pull-down data, PEAKS Studio software (Version 10.6, Bioinformatics Solutions Inc., Waterloo, Canada), in combination with a *Drosophila*-specific SwissProt database (UP000000803, www.uniprot.org/proteomes/UP000000803) was used to determine peptide-specific amino acid sequences (parent mass error tolerance: 10 ppm; fragment mass error tolerance: 0.2 Da; enzyme: trypsin; max missed cleavages: 2; selected PTMs: carbamidomethylation, oxidation, phosphorylation). Label-free quantification was performed by comparing peptide and protein amounts of different groups according to established protocols[64], with each group consisting of at least three independent biological replicates. The protein list was controlled by FDR (threshold: 1%) and significance was calculated by PEAKS software (one-way ANOVA). Classification as a possible interaction partner required $p < 0.01$, with quantification being based on at least two individual protein-specific peptides.

### Immunohistochemistry and Western blot
Animals were dissected on Sylgard plates (Sylgard 184 Elastomer Base and Curing Agent, Dow Corning, MI, USA), fixed in 3.7% formaldehyde in PBS for 1 h, rinsed three times in PBS, and transferred into 1.5 ml reaction cups. Subsequently, tissues were permeabilized in 1% Triton X-100 for 1 h, blocked in ROTI ImmunoBlock (45 min), and incubated with primary antibodies (overnight, 4 °C). Samples were washed in PBT (3×, 10 min each) and blocked again as described above. Secondary antibodies were applied for 90 min. For staining of human tissue, paraffin-embedded myocardial left ventricular heart slices (5 µm) were deparaffinized and rehydrated[65]. Tissue permeabilization was done in 0.1% Triton X-100 (15 min), followed by washing in PBS (3x) and blocking in 5% BSA. Primary antibodies were diluted in 5% BSA in PBS and incubated overnight. Sections were washed in 1% BSA in PBS (3×, 5 min each), followed by incubation with secondary antibodies (1 h, 4 °C). Finally, samples were washed as described above and mounted in Fluoromount-G. Primary antibodies used were: anti-Nep4 (RRID:AB_2569115, 1:200, monospecificity confirmed in ref. 28), anti-SERCA (1:500, *Drosophila*-specific, kind gift from Mani Ramaswami, monospecificity confirmed in ref. 66), anti-SERCA2 (human, ab219173, 1:100, Abcam, Cambridge, UK), anti-NEP (human, ab256494, 1:100, Abcam), anti-GFP (RRID:AB_889471, 1:500), anti-GFP (RRID:AB_305564, 1:2000), anti-GFP (RRID:AB_300798, 1:750), and anti-HA (RRID:AB_262051, 1:100). The secondary antibodies were anti-mouse-A647 (RRID:AB_2687948, 1:200), anti-mouse-Cy2 (RRID:AB_2307343, 1:100), anti-mouse-Cy3 (RRID:AB_2338680, 1:200), anti-rabbit-Cy2 (RRID:AB_2338021, 1:100), anti-rabbit-Cy3 (RRID:AB_2338000, 1:200), anti-rabbit-Dy550 (RRID:AB_10674190, 1:200), anti-goat-A488 (ab150141, Abcam, 1:150), and anti-chicken-A488 (RRID:AB_2340375,1:100). For STED imaging, Star 488 (2-0102-006-7, Abberior, Göttingen, Germany), Star Orange (RRID:AB_2847853), and Star Red (RRID:AB_2620152) secondary antibodies were used. Confocal images were captured with an LSM5 Pascal confocal microscope (Zeiss) using ZEN software (version 2.6, blue edition, Zeiss). STED imaging was performed using a Leica TCS STED CW microscope (Leica Microsystems, Wetzlar, Germany) and Leica Application Suite X software (version 3.5.2, LasX).

Western blots were performed according to standard protocols[25]. Primary antibodies were anti-Actin (RRID:AB_528068, 1:50), anti-Calnexin (RRID:AB_2722011, 1:500), anti-HA (RRID:AB_262051, 1:1000), anti-Nep4 (RRID:AB_2569115, 1:2000), anti-NEP (human, ab256494, 1:2000, Abcam), anti-SERCA (ref. 66, 1:5000), and anti-Calmodulin (RRID:AB_309644, 1:2000). Secondary antibodies were anti-mouse-alkaline phosphatase (AP) (RRID:AB_258091, 1:10,000) and anti-rabbit-alkaline phosphatase (RRID:AB_258446, 1:10.000). For quantification, pixel intensities of respective bands were measured using Image Lab Software, Version 6.0.1 (BioRad).

### Protein orientation
Determination of Nep4 orientation in SR membranes was performed according to ref. 34. In detail, the dorsal muscles of Nep4::roGFP2-expressing third instar larvae were dissected as described for the "analysis of cardiac function" (see above). As distinct from the heart preparations, the dorsal vessel as well as the main tracheal branches were also removed. Preparations were placed with the ventral side down on an object slide and covered with PBS. Confocal images were captured with an inverse LSM Olympus IX81 (Olympus, Tokyo, Japan) using FV1000 software (version 4.2.1.20, Olympus) and excitation at 405 nm and 488 nm. After measuring an untreated steady state, the PBS was replaced by 0.5 mM diamide, and the oxidized state was captured 1 min later. Subsequently, the diamide solution was replaced by 2 mM DTT and the reduced state was captured after 7 min of incubation. By calculating the ratio of the average 405 nm/488 nm signal intensities and comparing the steady-state ratio with the corresponding oxidized and reduced states, the redox environment of Nep4::roGFP in the untreated muscles was deduced. As a control, cytoplasmic roGFP2 was analyzed analogously.

### Statistics and reproducibility
For statistical analysis of physiological heart parameters, cardiac $Ca^{2+}$ transients, muscle performance, and Nep4 orientation in the SR membrane, one-way ANOVA followed by Dunnett's Multiple Comparison Test was used. Western blot quantifications and analyses of the effects of variable heart rates on $Ca^{2+}$ flux parameters (Supplementary Table 2) were statistically analyzed using a paired $t$ test (two-tailed) allowing pairing of related values from one individual replicate. For all tests, a $p$ value <0.05 was considered significant (*$p < 0.05$, **$p < 0.01$, ***$p < 0.001$). For all boxplots, the center line of a plot indicates the median; the upper and lower bounds indicate the 75th and 25th percentiles, respectively; and the whiskers indicate the minimum and maximum. Except for the pull-down data, which were statistically analyzed and visualized with PEAKS Studio software, all other data were analyzed and visualized using GraphPad Prism 5 (version 5.03, GraphPad Software Inc., San Diego, CA, USA). All stainings and Western blots were performed at least in triplicates.

### Reporting summary
Further information on research design is available in the Nature Research Reporting Summary linked to this article.

## Data availability
The data supporting the findings from this study are available within the manuscript and its supplementary information. The mass spectrometry proteomics data have been deposited to the ProteomeXchange Consortium via the PRIDE [67] partner repository (http://www.ebi.ac.uk/pride) with the dataset identifier PXD027738. SwissProt database (UP000000803, www.uniprot.org/proteomes/

UP000000803) was used to determine peptide-specific amino acid sequences. Source data are provided with this paper.

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

## Acknowledgements

We thank Martina Biedermann and Mechthild Krabusch for excellent technical assistance with fly work; Masaki Fukata, Yuko Fukata, Hirokazu Ishii, Tomomi Nemoto, and Mikio Furuse for providing expertise on STED microscopy; Michael Stuke for contributing to the protein orientation assay; Matthew Wolf for providing transgenic GCaMP3 stocks; and Jose Pueyo-Marques for sharing transgenic Sarcolamban stocks. Anti-SERCA antibodies were kindly provided by Mani Ramaswami. We further acknowledge the Vienna *Drosophila* Resource Center (VDRC) and the Bloomington *Drosophila* Stock Center (BDSC, NIH P40OD018537) for providing fly stocks. This research was funded by the German Research Foundation (SFB 944: Physiology and Dynamics of Cellular Microcompartments) (H. Meyer, A.P.), and by a stipend from the Hans Mühlenhoff Foundation (to R.S.). We also acknowledge the support of the Open Access Publishing Fund of Osnabrück University.

## Author contributions

R.S., A.B.: acquisition of data, analysis and interpretation of data, drafting or revising the article; E.C., P.F.: acquisition of data, drafting or revising the article; S.W., J.J.H., H. Milting, A.P.: analysis and interpretation of data, drafting or revising the article; H. Meyer: conception and design, acquisition of data, analysis and interpretation of data, drafting or revising the article.

## Funding

## Competing interests

The authors declare no competing interests.
