## [Peer Review File · Nature Communications]

Nepriylsins regulate muscle contraction and heart function via cleavage of SERCA-inhibitory micropeptidesReviewers' comments:

Reviewer #1 (Remarks to the Author):

The authors utilize drosophila models to demonstrate that increased expression of the endopeptidase Nep4 has increased incidence of cardiac arrhythmia. They attribute this observation to Nep4 cleavage of SCLA and SCLB and a corresponding increase in SERCA activity. However, this is only somewhat consistent with the data, as the authors make significant flaws in experimental design and make major logical leaps that preclude such conclusions. The authors do not show that Nep4 increases SERCA activity, which is essential, as SERCA's relationship to arrhythmia is controversial. In fact, it is more commonly observed that increased SERCA activity reduces arrhythmia. Furthermore, the authors do not demonstrate whether in vitro cleavage of SCLA/SCLB by Nep4 reflects something that occurs in intact drosophila. In addition, the authors fail to adequately describe their methods. Specific comments are listed below:

1) The authors do not describe Ca measurement methodology or results in sufficient detail. (a) The excitation/emission settings, imaging dimensions, recording/scan frequency for Ca measurements are not included in the text or references provided. (b) The concentration of caffeine used to estimate Ca load is not indicated. (c) It is unclear whether Ca measurements were made with paced or spontaneously beating samples (and if paced, the frequency and stimulation parameters used should be included). (d) While dot plots are included in Fig. 2, "n" number should be included in the figure legend as well. Presumably, the sample number corresponds to the number of animals, but this is not indicated. (e) Example recording traces should be included in Fig. 2.

2) GCaMP3 is an intensity-based fluorescent Ca indicator, which allows for assessment of relative changes within the same sample (e.g., same sample at different pacing frequencies or +/- a drug). These relative changes can then be compared between different samples. In order to make amplitude comparisons between samples (e.g., caffeine/Ca load) with an intensity based indicator, one must first normalize the signal to known Ca concentrations (typically high and low Ca) to account for variations in Ca indicator expression/loading, photobleaching, etc. The authors do not appear to have done so, making conclusions drawn from Fig. 2A suspect. Only ratiometric Ca indicators can be used without such normalization.

3) The authors do not describe methodology used to quantify SERCA activity in Fig. 2B. Presumably, they are estimating SERCA activity from Ca transient decays, but why does data in Fig. 2B differs from Fig. 2C (τ)? No calculation procedures are provided.

A further complication is that plasma membrane and mitochondrial Ca transport can also contribute to Ca transient decays (which don't appear to be accounted for) and it is unclear if samples were paced (related to Comment #1), as variations in beating frequencies (especially those seen in Fig. 1B) can significantly alter Ca flux parameters. As increased SERCA activity is central the conclusions, a more robust and direct measurement of activity is necessary. Specifically, the authors should co-transfect SERCA with SCLA/SCLB +/- Nep4 and perform an enzyme activity assay as is the standard for SERCA research.

4) Methods and equipment used for mass-spec analysis should be described.

5) The authors demonstrate that Nep4 can cleave peptide sequences of SCLA, SCLB, and SLN and that the corresponding truncations disrupt membrane localization. However, this experimental approach is insufficient to support the conclusions. The authors do not demonstrate that Nep4 can cleave full-length and membrane-inserted SCLA, SCLB, and SLN. Fig. 5 only demonstrates cleavage of short peptides in solution, and Fig. 6 only demonstrates the inability of newly translated proteins with truncations to insert into the membrane, not the ability of the protein being released from the membrane after truncation. This is a critical distinction, as the authors acknowledge that the catalytic activity of Nep4 is on the luminal membrane side, and thus any cleavage would only occur after membrane insertion of a fully translated protein. To support their mechanistic conclusions, authors would need to co-express Nep4 with full-length SCLA, SCLB, and SLN and observe loss of membrane localization. Most importantly, the authors would also need to demonstrate that animals overexpressing catalytically active Nep4 have reduced expression and/or membrane localization of SCLA and SCLB. If SCLA/SCLB expression and localization remained unchanged, any in vitro cleavage data would be moot and would not explain arrhythmia or potential increase in SERCA

activity.

Reviewer #2 (Remarks to the Author):

The authors report on the role of neprilysin in generating peptidic cleavage products that may regulate the activity of Ca²⁺ transports in the heart (here: Drosophila muscle).

This reviewer focuses on the mass spectrometric part. Generally, the mass spectrometric experiments appear to have been thoroughly conducted and reported. They predominantly serve as a "screening platform" with results being corroborated by further methods.

The mass spectrometric data should be uploaded to a repository such as PRIDE as it is standard in the field. It could be that in the MatMet section some "micro" characters have been accidentally converted to an "m"; please check.

It remains unclear how many replicates have been performed. Note that even a unique experiment might be fine for publication since (see above) results are corroborated by further methods.

Sample preparation for the cleavage product assays should be better described.

Please elaborate more on the type of database and search settings.

The authors state CID fragmentation. Wouldn't HCD be more correct for a Q-Exactive?

Fig 3G: Is the p-value FDR-controlled?

Generally, it would be good to see detection of the peptidic fragments in vivo, although this will prove to be very challenging.

Reviewer #3 (Remarks to the Author):

The manuscript of Schiemann et al provide for the first time evidence that neprilysins cleave SERCA-inhibitory micropeptides. Although it is well known that neprilysins cleave various peptides, so far it has not been described that also peptides such as Drosophila sarcolamban A/B and human sarcolipin represent a substrate for this protein. The authors provide some evidence that as such neprilysins regulate SR calcium homeostasis in Drosophila. Also, they provide some proof that indeed overexpression of NEP4 causes arrhythmicity and mutant NEP4 or suppression of NEP4 prevent these changes. Furthermore, localization of NEP4, SERCA and truncated peptides has been shown in Drosophila cells and NEP and SERCA localization in human ventricular cardiomyocytes.

The provided data is partly robust. The localization studies are clear, but the functional data can be improved.

1) Fig 1: overexpression of NEP4 causes arrhythmicity. This finding is based on only 5 data points. It seems that OE of NEP4 has an effect on part of the flies as N=5 flies do not show any effect. I suggest to confirm overexpression (suppression) of NEP4 in the individual flies and determine the sarcolamban levels as well. Same holds true for heart rate. In the manuscript has been mentioned that the OE flies are morphologically smaller compared to the other groups. So is effect on arrhythmia an artifact or real? Fig 1D shows arrhythmicity in siRNA NEP4 line, but this is not observed in Fig 1A. In Fig E and F you measure crawling speed in pupae. I'm not sure how this data relates to heart function and compared to the additional findings in the paper, body wall muscles are not discussed. In the text line: 126-128: '... catalytic activity and thus on impaired cleavage of distinct heart

regulatory peptides'. Please show cleaved peptides in the this experimental setup. Otherwise, this is an overstatement.

Line 153-154: 'increased physiological relevance...NEP4 activity'. As you cannot compare effect in pupae to effects in adult hearts this is an overstatement. Also, NEP4 activity has not been measured. Also this comparison; pupae muscle movement with adult fly heart wall measurements is not clear to me.

2) Fig 2: please show CaT tracers of the adult heart of Drosophila. Also in this figure, the OE NEP4 Dros show partly an effect: 4 samples show increased Ca²⁺ load and 4 samples are comparable to controls. Why? confirmation of NEP4 levels may provide information.

The description of the SERCA activity measurement is not very clear. Please provide a more detailed description. Fig 2D: add data on NEP4 levels (and also peptide expression levels).

3) The validity of data will improve by adding control data to supplemental sections. In the text, at several positions 'data not shown' is mentioned. For the transparency of the conducted experiments, I suggest to add the data to supplemental section.

4) Nep co-localized with SERCA and SCL. Interestingly this co-localization has especially been observed around the nucleus. I was wondering why a nuclear staining has been omitted. The nucleus can be easily stained with DAPI. At further distances from the nucleus only co-staining of SERCA and SCL has been observed (so no NEP4). Is this observation in line with the SR calcium measurements? As NEP4 is not present at SR localized at the sarcomeres, the effect on heart contraction may be limited and not the main mechanism of NEP4 in the cell and on SR function. This suggestion is in line with a partially effect of NEP4 OE on calcium flux and heart wall movements as observed in Fig 1 and 2.

But, the observation that NEP4 co-localizes with SCL and SERCA at the peri-nuclear area is interesting. Also the data showing NEP4 to cleave SCL and human SLN is striking. But the functional relevance is lacking. What will be the effect of truncated proteins in the nucleus or ER? Is the role of truncated SCL different in ER and SR?

5) Finally Dros data is expanded to human ventricular cardiomyocytes. Also here, SERCA and NEP are stained and show co-localization along the Z-discs and nucleus. SLN/PLN staining is missing. Also functional effects are missing. The data will significantly improve by adding CaT data, SERCA activity measurements and testing of specific NEP inhibitors.

6) For me the data describing the fluorescence intensity ratio is difficult to follow. This may be due to lack of background knowledge of this reviewer.

Overall, while the observation that NEP4 cleaves SCL/SLN in Dros cells, and NEP4 co-localizes with SERCA and SCL especially in Dros nearby the nucleus is of interest, the functional consequences are less clear. A partially effect on calcium homeostasis has been observed but an ER/nuclear role of NEP4 and truncated peptides is lacking. Also confirmation in human cardiomyocytes is only descriptive and without functional proof.

In addition, this manuscripts lacks some controls as mentioned above: W-blot showing NEP/SCL/SERCA expression levels in Dros (and human) samples, CaT tracers, NEP inhibitors and detailed description of methods related to CaT/SERCA activity measurements. The rationale to measure body movements in pupae is not clear to this reviewer.

Reviewer #1 (Remarks to the Author):

The authors utilize drosophila models to demonstrate that increased expression of the endopeptidase Nep4 has increased incidence of cardiac arrhythmia. They attribute this observation to Nep4 cleavage of SCLA and SCLB and a corresponding increase in SERCA activity. However, this is only somewhat consistent with the data, as the authors make significant flaws in experimental design and make major logical leaps that preclude such conclusions. The authors do not show that Nep4 increases SERCA activity, which is essential, as SERCA's relationship to arrhythmia is controversial. In fact, it is more commonly observed that increased SERCA activity reduces arrhythmia. Furthermore, the authors do not demonstrate whether in vitro cleavage of SCLA/SCLB by Nep4 reflects something that occurs in intact drosophila. In addition, the authors fail to adequately describe their methods. Specific comments are listed below:

1) The authors do not describe Ca measurement methodology or results in sufficient detail. (a) The excitation/emission settings, imaging dimensions, recording/scan frequency for Ca measurements are not included in the text or references provided.

Corresponding information have been added to the Materials and Methods section.

(b) The concentration of caffeine used to estimate Ca load is not indicated.

The concentration used (10 mM) has been included into the Materials and Methods section

(c) It is unclear whether Ca measurements were made with paced or spontaneously beating samples and if paced, the frequency and stimulation parameters used should be included).

Hearts were not paced and cardiac parameters were analyzed under the premise that the spontaneous heart rates of semi-intact preparations were statistically not significantly different between the individual groups of flies. Still, as also stated by the reviewer further below, there is some variability among the heart rates of flies of all analyzed genotypes. This "intrinsic" variability has been observed in *Drosophila* several times, yet it rarely turned out to be of statistical significance.

However, to exclude any effects caused by such variable beating frequencies, we repeated our analyses with a subgroup of *nep4* knockdown flies selected specifically for slower heart rate. As depicted in a novel Supplemental Table 2, for this group all effects on SR Ca²⁺ load, SERCA activity, and Tau were consistent with the population data for that genotype (Tab. S2, Fig. 2A-C), indicating that variations in heart rate do not significantly affect the measured parameters.

Corresponding information have been included into the Results and the Materials and Methods sections.

(d) While dot plots are included in Fig. 2, "n" number should be included in the figure legend as well. Presumably, the sample number corresponds to the number of animals, but this is not indicated.

The sample number indeed reflects the number of animals. All figure legends have been amended accordingly.

(e) Example recording traces should be included in Fig. 2.

Representative recording traces have been included into Fig. 2.

2) GCaMP3 is an intensity-based fluorescent Ca indicator, which allows for assessment of relative changes within the same sample (e.g., same sample at different pacing frequencies or +/- a drug). These relative changes can then be compared between different samples. In order to make amplitude comparisons between samples (e.g., caffeine/Ca load) with an intensity based indicator, one must first normalize the signal to known Ca concentrations (typically high and low Ca) to account for variations in Ca indicator expression/loading, photobleaching, etc. The authors do not appear to have done so, making conclusions drawn from Fig. 2A suspect. Only ratiometric Ca indicators can be used without such normalization.

Normalization was done. To correct for variable indicator concentration, we normalized using the formula " $F-F_0/F_0$ ". In this regard, F represents the maximal fluorescence of each peak (systolic) and F_0 the minimal fluorescence (diastolic). This approach normalizes differences in indicator concentration between cells, thus providing a plausible method for comparing ratios between different samples / genotypes. Based on this approach, numerous analyses have been conducted in *Drosophila* (e.g. ^{1,2,3}).

Materials and Methods section was amended to include these information.

3) The authors do not describe methodology used to quantify SERCA activity in Fig. 2B. Presumably, they are estimating SERCA activity from Ca transient decays, but why does data in Fig. 2B differs from Fig. 2C (tau)? No calculation procedures are provided.

A detailed description of the methodology to quantify SERCA activity as well as calculation procedures have now been included into the Materials and Methods section. We apologize for omitting these essential information in the original manuscript.

By providing the calculation procedures we can also clarify the apparent discrepancy between Fig. 2B and 2C. As described in more detail below as well as in the amended Materials and Methods section, in contrast to Fig. 2B that exclusively depicts calculated SERCA activities, Fig. 2C (tau) reflects the combined activities of SERCA and of the Na^+ / Ca^{2+} exchanger (NCX). Thus, a significant effect in Fig. 2B (significant for Nep4-RNAi and Nep4 overexpression) does not necessarily premise significance in Fig. 2C (significant for Nep4-RNAi, tendency for Nep4 overexpression).

The inverse relationship between Fig 2B and 2C is explained by the fact that Fig. 2C depicts tau as a time constant that represents the decay of Ca^{2+} transients in the untreated state (without caffeine) and not enzyme activities. Accordingly, tau is measured, and also commonly illustrated, in [sec] or [msec]. On the other hand, we depicted SERCA activity as [1/sec], which explains the at first sight contrary appearances of the values in Figs. 2B and 2C.

We included detailed calculation procedures into the amended Material and Methods section to clarify the issue.

A further complication is that plasma membrane and mitochondrial Ca transport can also contribute to Ca transient decays (which don't appear to be accounted for) and it is unclear if samples were paced (related to Comment #1), as variations in beating frequencies (especially those seen in Fig. 1B) can significantly alter Ca flux parameters. As increased SERCA activity is central the conclusions, a more robust and direct measurement of activity is necessary. Specifically, the authors should co-transfect SERCA with SCLA/SCLB +/- Nep4 and perform an enzyme activity assay as is the standard for SERCA research.

For cardiac relaxation to occur, Ca^{2+} is quickly removed from the cytosol, essentially by the action of SERCA and NCX, which together account for 98-99% of the total Ca^{2+} decay (4). The amount of Ca^{2+} that leaves the cytosol via the remaining routes, i.e. slow removal systems such as the mitochondrial Ca^{2+} uniporter or the sarcolemmal Ca^{2+} -ATPase, is rather inconsequential with respect to excitation-contraction coupling (ECC, 4, 5, 6). Since ECC in flies and mammals has been shown to be highly similar, with all mammalian ECC protein landmarks being present in *Drosophila*, comparable Ca^{2+} handling rates are assumed in fly hearts (7, 8, 9, 10, 11, 12).

To discriminate between SERCA and NCX activity, we analyzed caffeine-induced Ca^{2+} -transients and subtracted the inverse of the rate constants of caffeine induced Ca^{2+} transients decays (Tau_{caff}) from that of the pre-caffeine transients (Tau_{ca}), ($1/\text{Tau}_{\text{ca}} - 1/\text{Tau}_{\text{caff}}$). In this regard, the rate constant of $[\text{Ca}^{2+}]_i$ decay during a caffeine-induced Ca^{2+} transient largely reflects the function of NCX, while both NCX and SERCA contribute to the pre-caffeine $[\text{Ca}^{2+}]_i$ decay. This approach to calculate SERCA activity is widely accepted in the field and has been applied in numerous studies (e.g. 2, 13, 14, 15, 16, 17, 18, 19, 20).

The Materials and Methods section has been amended to include these information.

To exclude any influence of variable beating frequencies on the measured Ca^{2+} flux parameters, we analysed a subgroup of *nep4* knockdown flies selected specifically for slower heart rates. As depicted in a novel Supplemental Table 2, for this group all effects on SR Ca^{2+} load, SERCA activity, and Tau were consistent with the population data (Tab. S2, Fig. 2A-C), indicating that variations in heart rate do not significantly affect the measured parameters.

Corresponding information have been included into the Results section.

4) Methods and equipment used for mass-spec analysis should be described.

The Materials and Methods section was amended accordingly.

The mass spectrometric dataset has been uploaded to "ProteomeXchange" via the PRIDE database, as also suggested by reviewer 2. It can be accessed here (<http://www.ebi.ac.uk/pride>) using the following login details:

Username: reviewer_pxd027738@ebi.ac.uk

Password: 6nQ54sUQ

5) The authors demonstrate that Nep4 can cleave peptide sequences of SCLA, SCLB, and SLN and that the corresponding truncations disrupt membrane localization. However, this experimental approach is insufficient to support the conclusions. The authors do not demonstrate that Nep4 can cleave full-length and membrane-inserted SCLA, SCLB, and SLN. Fig. 5 only demonstrates cleavage of short peptides in solution, and Fig. 6 only demonstrates the inability of newly translated proteins with truncations to insert into the membrane, not the ability of the protein being released from the membrane after truncation. This is a critical distinction, as the authors acknowledge that the catalytic activity of Nep4 is on the luminal membrane side, and thus any cleavage would only occur after membrane insertion of a fully translated protein. To support their mechanistic conclusions, authors would need to co-express Nep4 with full-length SCLA, SCLB, and SLN and observe loss of membrane localization. Most importantly, the authors would also need to demonstrate that animals overexpressing catalytically active Nep4 have reduced expression and/or membrane localization of

SCLA and SCLB. If SCLA/SCLB expression and localization remained unchanged, any *in vitro* cleavage data would be moot and would not explain arrhythmia or potential increase in SERCA activity.

We performed the experiments as suggested by the reviewer and found that co-expression of Nep4 (or human NEP) together with the individual SERCA regulatory peptides significantly reduced membrane localization of the peptides in S2 cells. Without Nep4 (or human NEP), the same peptides were largely membrane bound. These results provide strong support for our mechanistic model that the neprilysin-mediated cleavage affects membrane anchoring of the peptides, thereby releasing them from the SR membrane and preventing abnormal peptide accumulation and excessive SERCA inhibition.

In transgenic animals, loss of membrane localization was not clearly observable, probably due to an efficient degradation of the soluble Nep4-cleaved peptides in the cytoplasm of muscle cells, which may not occur that efficiently in S2 cells. We included this possibility along with information on the instability and rapid intracellular degradation of truncated forms of vertebrate PLN²¹ and SLN²² into the "Discussion" section.

Strikingly, we found that in the same transgenic animals the overall amount of SCLA was considerably reduced if Nep4 was co-expressed with the peptide (new Fig. 6D, E). In addition, SCLA and SCLB formed high molecular weight species, probably representing peptide oligomers, which were largely absent if Nep4 was co-expressed (new Fig. 6D, E). This is a remarkable observation considering that oligomerization is also a well-known and physiologically highly relevant characteristic of vertebrate Phospholamban²³. To our knowledge, this is the first indication of an ability of SCL peptides to form oligomers.

From these results we concluded that i) Nep4 hydrolyzes SCLA and SCLB *in vivo*, ii) this cleavage event reduces membrane localization of both peptides as well as their ability to oligomerize, and iii) cleavage considerably reduces the overall amount of SCLA in muscle tissue. These conclusions are consistent with analyses on human PLN that confirmed for the C-terminally truncated peptide i) a reduced membrane localization, ii) a reduced ability to form oligomers, and iii) a reduced binding affinity to SERCA²⁴. Thus, in addition to confirming the *in vivo* relevance of the Nep4 mediated SCL cleavage, this set of experiments further corroborated the indication that *Drosophila* and vertebrate SERCA-inhibitory peptides act in a mechanistically similar manner.

The new datasets showing the effects of co-expressing Nep4 together with the SERCA regulatory peptides in S2 cells and in transgenic animals are depicted in a novel Fig. 6. The individual molecular consequences observed for Nep4-mediated SCLA and SCLB cleavage, as well as for S2 cells and transgenic animals, are addressed in an amended "Discussion" section.

Reviewer #2 (Remarks to the Author):

The authors report on the role of neprilysin in generating peptidic cleavage products that may regulate the activity of Ca²⁺ transports in the heart (here: *Drosophila* muscle).

This reviewer focuses on the mass spectrometric part. Generally, the mass spectrometric experiments appear to have been thoroughly conducted and reported. They predominantly serve as a "screening platform" with results being corroborated by further methods.

The mass spectrometric data should be uploaded to a repository such as PRIDE as it is standard in the field.

The mass spectrometric dataset has been uploaded to "ProteomeXchange" via the PRIDE database as suggested. It can be accessed here (<http://www.ebi.ac.uk/pride>) using the following login details:

Username: reviewer_pxd027738@ebi.ac.uk

Password: 6nQ54sUQ

It could be that in the MatMet section some "micro" characters have been accidentally converted to an "m"; please check.

We thank the reviewer for recognizing this mistake. The respective characters have been replaced.

It remains unclear how many replicates have been performed. Note that even a unique experiment might be fine for publication since (see above) results are corroborated by further methods.

For each mass spectrometric experiment, at least three individual replicates were performed. This information has now been included into the Materials and Methods section.

Sample preparation for the cleavage product assays should be better described. Please elaborate more on the type of database and search settings.

The Materials and Methods section was amended accordingly.

The authors state CID fragmentation. Wouldn't HCD be more correct for a Q-Exactive?

The reviewer is right and we apologize for the mix-up. We amended the Materials and Methods section accordingly.

Fig 3G: Is the p-value FDR-controlled?

Yes; this information has now been included into the Materials and Methods section.

Generally, it would be good to see detection of the peptidic fragments *in vivo*, although this will prove to be very challenging.

We indeed tried to detect the peptidic fragments resulting from Nep4-mediated Sarcolamban-cleavage *in vivo* but were not successful. In addition to the possible technical reasons, this could also reflect efficient intracellular degradation of the cleaved peptides, as also discussed above.

Reviewer #3 (Remarks to the Author):

The manuscript of Schiemann et al provide for the first time evidence that neprilysins cleave SERCA-inhibitory micropeptides. Although it is well known that neprilysins cleave various peptides, so far it has not been described that also peptides such as Drosophila sarcolamban A/B and human sarcolipin represent a substrate for this protein. The authors provide some evidence that as such neprilysins regulate SR calcium homeostasis in Drosophila. Also, they provide some proof that indeed overexpression of NEP4 causes arrhythmicity and mutant NEP4 or suppression of NEP4 prevent these changes. Furthermore, localization of NEP4, SERCA and truncated peptides has been shown in Drosophila cells and NEP and SERCA localization in human ventricular cardiomyocytes.

The provided data is partly robust. The localization studies are clear, but the functional data can be improved.

1) Fig 1: overexpression of NEP4 causes arrhythmicity. This finding is based on only 5 data points. It seems that OE of NEP4 has an effect on part of the flies as N=5 flies do not show any effect. I suggest to confirm overexpression (suppression) of NEP4 in the individual flies and determine the sarcolamban levels as well. Same holds true for heart rate.

As suggested, we analyzed the Nep4 overexpression efficiency in cardiomyocytes via immunocytochemistry. In comparison to controls, Nep4 overexpression animals exhibited a strong Nep4 signal. Within the individual Nep4 overexpression animals, we did not observe any clear differences in staining intensities. Thus, the individual-specific effects are apparently not caused by variable Nep4 levels.

Respective stainings are now shown in a novel Fig. S9A, and explanatory information have been included into the Results section.

Of note, similar inter-individual differences have also been observed for temperature-sensitive *Drosophila* SERCA mutants (*CaP60A*^{Kum170},²⁵). In that case, incubation at restrictive temperatures affected rhythmicity of only some animals, while others were not affected. Thus, inter-individual differences appear to be a general phenomenon associated with cardiac rhythmicity in *Drosophila*.

Due to lack of specific Sarcolamban antibodies, corresponding stainings could not be performed. Anti-Sarcolipin antibodies did not detect Sarcolamban peptides in our hands. This was unexpected, because an earlier publication⁽¹¹⁾ stated that antibodies to human Sarcolipin recognize Sarcolamban A and B.

In the manuscript has been mentioned that the OE flies are morphologically smaller compared to the other groups. So is effect on arrhythmia an artifact or real?

The effect of a reduced body size was observed exclusively in animals that overexpressed Nep4 in all muscle cells, including the body wall muscles (*mef2*-Gal4 driver). Corresponding animals were analyzed only for aberrant crawling abilities, with the reduced size taken into account (Fig. 1E, F).

By contrast, heart parameters were measured exclusively in animals with cardiac-specific overexpression of Nep4 (*tinC*-Gal4 driver), and such animals did not exhibit any noticeable alterations in size.

Fig 1D shows arrhythmicity in siRNA NEP4 line, but this is not observed in Fig 1A.

In the course of our analyses, we also got the impression that *nep4* knockdown hearts exhibited a tendency towards arrhythmia. However, the impairments were not statistically significant (Fig. 1A). To still visualize these effects, we decided to show combined histograms of the distribution of heart periods in animals of the individual genotypes. In these histograms, abnormally long heart periods became visible in *nep4* RNAi hearts (Fig. 1C, arrows), and these abnormal heart periods are also depicted in Fig 1D.

The occurrence of abnormal heart periods in *nep4* RNAi hearts, as well as the lack of statistical significance, is now mentioned in the Results section.

In Fig E and F you measure crawling speed in pupae. I'm not sure how this data relates to heart function and compared to the additional findings in the paper, body wall muscles are not discussed.

It is true that this paper focuses on heart function and on the physiological effects caused by variable Nep4 activities. However, we consider the observation that the regulatory mechanism we describe for cardiomyocytes appears to be present also in body wall muscles relevant. This holds true especially in light of the ongoing identification of SERCA-regulatory micropeptides in tissues other than the heart (e.g. Myoregulin in skeletal muscle or Endoregulin in nonmuscle cell types). Also for these peptides, mechanisms regulating their turnover / homeostasis are largely unknown and neprilysin activity may prove to be essential here as well. Possible phylogenetic implications of this emerging tissue-independent functional conservation are discussed.

In the text line: 126-128: '... catalytic activity and thus on impaired cleavage of distinct heart regulatory peptides'. Please show cleaved peptides in the this experimental setup. Otherwise, this is an overstatement.

The respective statement was removed.

Line 153-154: 'increased physiological relevance...NEP4 activity'. As you cannot compare effect in pupae to effects in adult hearts this is an overstatement. Also, NEP4 activity has not been measured. Also this comparison; pupae muscle movement with adult fly heart wall measurements is not clear to me.

The statement was removed.

2) Fig 2: please show CaT tracers of the adult heart of *Drosophila*.

Representative recording traces have been included into Fig. 2.

Also in this figure, the OE NEP4 *Dros* show partly an effect: 4 samples show increased Ca²⁺ load and 4 samples are comparable to controls. Why? Confirmation of NEP4 levels may provide information.

As stated above, no differences in Nep4 levels were observed via immunostainings of corresponding overexpression animals. Yet, similar inter-individual differences have also been observed for temperature-sensitive *Drosophila* SERCA mutants (*CaP60A*^{Kum170, 25}), indicating that such differences represent a general phenomenon associated with cardiac Ca²⁺ handling in *Drosophila*.

The description of the SERCA activity measurement is not very clear. Please provide a more detailed description.

Corresponding information have been included into the Materials and Methods section.

Fig 2D: add data on NEP4 levels (and also peptide expression levels).

Nep4 levels in the corresponding lines have been described previously (²⁶). Citation has been included.

As stated above, due to lack of specific Sarcolamban antibodies, peptide expression levels could not be assessed.

3) The validity of data will improve by adding control data to supplemental sections. In the text, at

several positions 'data not shown' is mentioned. For the transparency of the conducted experiments, I suggest to add the data to supplemental section.

The control staining for Nep4::HA localization is now included in Fig 3. The additional control for the pull-down assays (SR-luminal GFP) as well as the dataset depicting the magnitude of body wall muscle contractions in different *nep4* expression backgrounds are now shown in a novel Fig. S9.

The mass spectrometric dataset has been uploaded to "ProteomeXchange" via the PRIDE database, as also suggested by reviewer 2. It can be accessed here (<http://www.ebi.ac.uk/pride>) using the following login details:

Username: reviewer_pxd027738@ebi.ac.uk

Password: 6nQ54sUQ

4) Nep co-localized with SERCA and SCL. Interestingly this co-localization has especially been observed around the nucleus. I was wondering why a nuclear staining has been omitted. The nucleus can be easily stained with DAPI. At further distances from the nucleus only co-staining of SERCA and SCL has been observed (so no NEP4). Is this observation in line with the SR calcium measurements? As NEP4 is not present at SR localized at the sarcomeres, the effect on heart contraction may be limited and not the main mechanism of NEP4 in the cell and on SR function. This suggestion is in line with a partially effect of NEP4 OE on calcium flux and heart wall movements as observed in Fig 1 and 2.

As suggested, we now included DAPI-based nuclear stainings into Figure 4.

Regarding Nep4 localization, our data indeed indicate that the protein is somewhat enriched in SR membranes proximal to the nucleus. However, this effect is most prominent in animals expressing Nep4 under the control of the muscle-specific *mef2*-enhancer (Fig. 4). At endogenous expression levels, the protein is also clearly present at the SR localized at the sarcomeres (Fig. 3A', D', arrows). Thus, we consider the differences between perinuclear Nep4 localization and Nep4 localization more distant from the nucleus not sufficiently distinct to deduce any physiological implications.

We attenuated our statements regarding the protein localization accordingly.

But, the observation that NEP4 co-localizes with SCL and SERCA at the peri-nuclear area is interesting. Also the data showing NEP4 to cleave SCL and human SLN is striking. But the functional relevance is lacking. What will be the effect of truncated proteins in the nucleus or ER? Is the role of truncated SCL different in ER and SR?

Based on our data, we consider the possibility of a distinct biological function of the truncated peptides unlikely. The cleavage event rather appears to be required to control homeostasis of the full-length SERCA regulatory micropeptides within the SR membrane. We could confirm that Nep4-mediated cleavage significantly reduces membrane localization of these peptides. Without cleavage, the same peptides were largely membrane bound (Fig. 6, S6). This clear effect substantially supports our mechanistic model that the Nep4-mediated cleavage reduces membrane anchoring of the peptides, thereby releasing them from the SR membrane and preventing peptide accumulation and a resultant abnormal increase in SERCA interaction / inhibition.

Of note, truncated forms of Phospholamban and Sarcolipin have been reported to be highly unstable and prone to rapid intracellular degradation (²¹, ²²). We consider a similar fate also likely for truncated Sarcolamban.

The new dataset showing the effects of co-expressing Nep4 / NEP together with the corresponding SERCA regulatory peptides is depicted in a novel Fig. 6.

Information on the instability of the truncated SERCA regulatory peptides has been included into the Discussion section.

5) Finally Dros data is expanded to human ventricular cardiomyocytes. Also here, SERCA and NEP are stained and show co-localization along the Z-discs and nucleus. SLN/PLN staining is missing. Also functional effects are missing. The data will significantly improve by adding CaT data, SERCA activity measurements and testing of specific NEP inhibitors.

We agree that functional data from the human system would further strengthen the manuscript. However, in-depth analyses on vertebrate systems are beyond the scope of the present paper, which introduces neprilysin activity as a novel means to regulate SERCA function in *Drosophila*. While our data (especially Figs. 5, 6, S6, and S8) clearly indicate that Neprilysin mediated regulation of SERCA activity is relevant in humans and that the underlying molecular mechanisms may be evolutionarily conserved, dedicated future studies are required to assess this issue in detail.

6) For me the data describing the fluorescence intensity ratio is difficult to follow. This may be due to lack of background knowledge of this reviewer.

We included a novel set of experiments that utilize subcellular fractionation in combination with Western blot-based signal intensity measurements as a means to quantify the extent of peptide re-localization that occurs in response to Nep4-mediated cleavage.

The corresponding dataset replaces the fluorescence intensity ratio measurements originally shown in Fig. 6 and is now depicted in a novel Fig. 6.

Overall, while the observation that NEP4 cleaves SCL/SLN in Dros cells, and NEP4 co-localizes with SERCA and SCL especially in Dros nearby the nucleus is of interest, the functional consequences are less clear. A partially effect on calcium homeostasis has been observed but an ER/nuclear role of NEP4 and truncated peptides is lacking. Also confirmation in human cardiomyocytes is only descriptive and without functional proof.

In addition, this manuscripts lacks some controls as mentioned above: W-blot showing NEP/SCL/SERCA expression levels in Dros (and human) samples, CaT tracers, NEP inhibitors and detailed description of methods related to CaT/SERCA activity measurements. The rationale to measure body movements in pupae is not clear to this reviewer.

We thank all reviewers for their helpful and constructive comments! Based on these remarks, we did a series of additional experiments and were able to address most of the issues raised. Thus, the present manuscript represents a substantially amended version of the original paper, and we are confident that it now complies with the standards of "Nature Communications".

References

1. Balcazar D, *et al.* SERCA is critical to control the Bowditch effect in the heart. *Sci Rep* **8**, 12447 (2018).
2. Gomez IM, *et al.* Inhalation of marijuana affects *Drosophila* heart function. *Biol Open* **8**, (2019).
3. Santalla M, *et al.* Aging and CaMKII alter intracellular Ca²⁺ transients and heart rhythm in *Drosophila melanogaster*. *PLoS One* **9**, e101871 (2014).
4. Bers DM. Cardiac excitation-contraction coupling. *Nature* **415**, 198-205 (2002).
5. Brini M, Carafoli E. The plasma membrane Ca²⁺ ATPase and the plasma membrane sodium calcium exchanger cooperate in the regulation of cell calcium. *Cold Spring Harb Perspect Biol* **3**, (2011).
6. Li L, Chu G, Kranias EG, Bers DM. Cardiac myocyte calcium transport in phospholamban knockout mouse: relaxation and endogenous CaMKII effects. *Am J Physiol* **274**, H1335-1347 (1998).
7. Abraham DM, Wolf MJ. Disruption of sarcoendoplasmic reticulum calcium ATPase function in *Drosophila* leads to cardiac dysfunction. *PLoS One* **8**, e77785 (2013).
8. Cammarato A, *et al.* A mighty small heart: the cardiac proteome of adult *Drosophila melanogaster*. *PLoS One* **6**, e18497 (2011).
9. Desai-Shah M, Papoy AR, Ward M, Cooper RL. Roles of the Sarcoplasmic/Endoplasmic Reticulum Ca²⁺-ATPase, Plasma Membrane Ca²⁺-ATPase and Na⁺/Ca²⁺ Exchanger in Regulation of Heart Rate in Larval *Drosophila*. *The Open Physiology Journal* **3**, 16-36 (2010).
10. Limpitikul WB, Viswanathan MC, O'Rourke B, Yue DT, Cammarato A. Conservation of cardiac L-type Ca²⁺ channels and their regulation in *Drosophila*: A novel genetically-pliable channelopathic model. *J Mol Cell Cardiol* **119**, 64-74 (2018).
11. Magny EG, *et al.* Conserved regulation of cardiac calcium uptake by peptides encoded in small open reading frames. *Science* **341**, 1116-1120 (2013).
12. Taghli-Lamalle O, Plantie E, Jagla K. *Drosophila* in the Heart of Understanding Cardiac Diseases: Modeling Channelopathies and Cardiomyopathies in the Fruitfly. *J Cardiovasc Dev Dis* **3**, (2016).
13. Bassani JW, Bassani RA, Bers DM. Relaxation in rabbit and rat cardiac cells: species-dependent differences in cellular mechanisms. *J Physiol* **476**, 279-293 (1994).

14. Cheng J, *et al.* CaMKII inhibition in heart failure, beneficial, harmful, or both. *Am J Physiol Heart Circ Physiol* **302**, H1454-1465 (2012).
15. Diaz ME, Graham HK, O'Neill S C, Trafford AW, Eisner DA. The control of sarcoplasmic reticulum Ca content in cardiac muscle. *Cell Calcium* **38**, 391-396 (2005).
16. Diaz ME, Graham HK, Trafford AW. Enhanced sarcolemmal Ca²⁺ efflux reduces sarcoplasmic reticulum Ca²⁺ content and systolic Ca²⁺ in cardiac hypertrophy. *Cardiovasc Res* **62**, 538-547 (2004).
17. Fakuade FE, *et al.* Altered Atrial Cytosolic Calcium Handling Contributes to the Development of Postoperative Atrial Fibrillation. *Cardiovasc Res*, (2020).
18. Gao J, *et al.* Assessment of Sarcoplasmic Reticulum Calcium Reserve and Intracellular Diastolic Calcium Removal in Isolated Ventricular Cardiomyocytes. *J Vis Exp*, (2017).
19. Mederle K, *et al.* The angiotensin receptor-associated protein Atrap is a stimulator of the cardiac Ca²⁺-ATPase SERCA2a. *Cardiovasc Res* **110**, 359-370 (2016).
20. Piacentino V, 3rd, *et al.* Cellular basis of abnormal calcium transients of failing human ventricular myocytes. *Circ Res* **92**, 651-658 (2003).
21. Haghighi K, *et al.* Human phospholamban null results in lethal dilated cardiomyopathy revealing a critical difference between mouse and human. *J Clin Invest* **111**, 869-876 (2003).
22. Gramolini AO, Kislinger T, Asahi M, Li W, Emili A, MacLennan DH. Sarcolipin retention in the endoplasmic reticulum depends on its C-terminal RSYQY sequence and its interaction with sarco(endo)plasmic Ca(2+)-ATPases. *Proc Natl Acad Sci U S A* **101**, 16807-16812 (2004).
23. MacLennan DH, Kranias EG. Phospholamban: a crucial regulator of cardiac contractility. *Nat Rev Mol Cell Biol* **4**, 566-577 (2003).
24. Abrol N, *et al.* Phospholamban C-terminal residues are critical determinants of the structure and function of the calcium ATPase regulatory complex. *J Biol Chem* **289**, 25855-25866 (2014).
25. Sanyal S, Jennings T, Dowse H, Ramaswami M. Conditional mutations in SERCA, the Sarco-endoplasmic reticulum Ca²⁺-ATPase, alter heart rate and rhythmicity in *Drosophila*. *J Comp Physiol B* **176**, 253-263 (2006).
26. Panz M, Vitos-Faleato J, Jendretzki A, Heinisch JJ, Paululat A, Meyer H. A novel role for the non-catalytic intracellular domain of Neprilysins in muscle physiology. *Biol Cell* **104**, 553-568 (2012).

REVIEWER COMMENTS

Reviewer #1 (Remarks to the Author):

In this revised manuscript, the authors do a commendable job thoroughly addressing comments from all reviewers, substantially strengthening their presented work. In particular, the newly performed and included cleavage experiments are especially noteworthy. However, data corresponding to the GCaMP3 calcium indicator is still problematic, and not adequately addressed given that the related calcium measurements are a core component of this study.

In the initial review comments, this reviewer apologizes for a poor word choice of “normalization” when “calibration” would have been more appropriate when discussing the limitations of the intensity-based indicators. While the normalization performed by the authors is not wrong per se, it is insufficient to address sample-to-sample variation. A proper calibration to low and high calcium standards is usually required for such an indicator (references below). Due to the centrality of calcium measurements to this study, such a robust and quantifiable approach is recommended.

<https://pubmed.ncbi.nlm.nih.gov/21575569/>

<https://pubmed.ncbi.nlm.nih.gov/21807862/>

<https://pubmed.ncbi.nlm.nih.gov/18848629/>

Reviewer #2 (Remarks to the Author):

All comments have been adequately addressed. I congratulate the authors on this interesting study!

Reviewer #3 (Remarks to the Author):

This reviewer wants to congratulate Schiemann et al with collecting such an interesting set of data showing that neprilysins cleave SERCA-inhibitory micropeptides and as such regulate SERCA activity and contractile function in muscle and cardiomyocytes.

Although it is well known that neprilysins cleave various peptides, so far it has not been described that also peptides such as *Drosophila* sarcolamban A/B and human sarcolipin represent a substrate for this protein. As SERCA plays a key role in calcium handling and as such muscle and cardiomyocyte function, novel insights in how neprilysins regulate this evolutionary conserved process is of importance.

In the current version of the manuscript, the authors provide new evidence substantiating the role of Nep4 on the cleavage of micropeptides. In this version, all my previous comments have been answered appropriately. Future research should elucidate the role of Nep4 in muscle and cardiac disease onset and progression. The findings may have important consequences on our understanding of these diseases.

Reviewer #1 (Remarks to the Author):

In this revised manuscript, the authors do a commendable job thoroughly addressing comments from all reviewers, substantially strengthening their presented work. In particular, the newly performed and included cleavage experiments are especially noteworthy. However, data corresponding to the GCaMP3 calcium indicator is still problematic, and not adequately addressed given that the related calcium measurements are a core component of this study.

In the initial review comments, this reviewer apologizes for a poor word choice of “normalization” when “calibration” would have been more appropriate when discussing the limitations of the intensity-based indicators. While the normalization performed by the authors is not wrong per se, it is insufficient to address sample-to-sample variation. A proper calibration to low and high calcium standards is usually required for such an indicator (references below). Due to the centrality of calcium measurements to this study, such a robust and quantifiable approach is recommended.

We agree that utilization of ratiometric indicators represents the gold standard for quantitative Ca²⁺ measurements under various conditions. However, since we always express the changes in fluorescence signal intensity relative to the previously measured animal-specific min/max (diastole/systole) intensities and normalize the individual signals to the corresponding basal fluorescence ($F_{max}-F_0/F_0$), we consider our approach to account for sample-to-sample variation still adequate. This way to present fractional fluorescence changes has been applied for single-wavelength Ca²⁺ indicators in several models (e.g. Cavagna et al, 2000; Hadad et al, 2013; Wu et al, 2015; Balcazar et al, 2018; Tamayo et al, 2020; Tester et al, 2020). Moreover, since we apply a genetically encoded Ca²⁺ indicator (GCaMP3), variable cell membrane penetration efficiencies and a resulting inhomogeneous indicator loading, which represent common drawbacks of chemical Ca²⁺ indicators, can be largely excluded in our system.

Yet, to provide additional support for the validity of our measurements and to assess possible sample-to-sample variations more directly, we did Western blot based single-animal specific quantifications of GCaMP3 amounts in cardiomyocytes of all transgenic lines tested. As depicted in an amended Fig. S9E, GCaMP3 protein levels were identical in all animals, thus at least minimizing the possibility of artificial effects caused by animal-specific indicator concentrations. In addition, this new set of experiments excluded the option that the individual genetic backgrounds of the analyzed animals affected indicator concentrations.

The Results section was amended to include these information.

Reviewer #2 (Remarks to the Author):

All comments have been adequately addressed. I congratulate the authors on this interesting study!

Reviewer #3 (Remarks to the Author):

This reviewer wants to congratulate Schiemann et al with collecting such an interesting set of data showing that neprilysins cleave SERCA-inhibitory micropeptides and as such regulate SERCA activity and contractile function in muscle and cardiomyocytes.

Although it is well known that neprilysins cleave various peptides, so far it has not been described that also peptides such as Drosophila sarcolamban A/B and human sarcolipin represent a substrate for this protein. As SERCA plays a key role in calcium handling and as such muscle and cardiomyocyte function, novel insights in how neprilysins regulate this evolutionary conserved process is of importance.

In the current version of the manuscript, the authors provide new evidence substantiating the role of Nep4 on the cleavage of micropeptides. In this version, all my previous comments have been answered appropriately. Future research should elucidate the role of Nep4 in muscle and cardiac disease onset and progression. The findings may have important consequences on our understanding of these diseases.

We thank all reviewers for their very helpful and constructive comments!

References

1. Cavagna M, *et al.* Exogenous Ca²⁺-ATPase isoform effects on Ca²⁺ transients of embryonic chicken and neonatal rat cardiac myocytes. *J Physiol* **528**, 53-63 (2000).
2. Hadad I, *et al.* Stroma Cell-Derived Factor-1a Signaling Enhances Calcium Transients and Beating Frequency in Rat Neonatal Cardiomyocytes. *Plos One* **8**(2): e56007 (2013).
3. Wu A, *et al.* Breadth of tuning in taste afferent neurons varies with stimulus strength. *Nat Commun* **16**(6): 8171 (2015).
4. Balcazar D, *et al.* SERCA is critical to control the Bowditch effect in the heart. *Sci Rep* **8**:12447 (2018).
5. Tamayo M, *et al.* Intracellular calcium mishandling leads to cardiac dysfunction and ventricular arrhythmias in a mouse model of propionic academia. *Biochim Biophys Acta Mol Basis Dis* **1**(1): 165586 (2020).
6. Tester DJ, *et al.* Molecular characterization of the calcium release channel deficiency syndrome. *JCI Insight* **5**(15):e135952 (2020).

REVIEWERS' COMMENTS

Reviewer #4 (Remarks to the Author):

The authors' finding that neprilysins cleave SERCA regulatory peptides is an exciting and novel contribution to the field. As the field of micropeptides continues to expand, this study may have broader implications. I believe that all comments have been adequately addressed. Whether or not the GCaMP3 calcium indicator is calibrated for calcium concentration is unlikely to change the findings of the study.

Response to Reviewers

Reviewer #4 (Remarks to the Author):

The authors' finding that neprilysins cleave SERCA regulatory peptides is an exciting and novel contribution to the field. As the field of micropeptides continues to expand, this study may have broader implications. I believe that all comments have been adequately addressed. Whether or not the GCaMP3 calcium indicator is calibrated for calcium concentration is unlikely to change the findings of the study.

We thank the reviewer for the willingness to review our paper and, of course, for the supportive comments!